# Joint-Embedding vs Reconstruction: Provable Benefits of Latent Space Prediction for Self-Supervised Learning

**Hugues Van Assel**[*1,2], **Mark Ibrahim**[3], **Tommaso Biancalani**[1],
**Aviv Regev**[1], **Randall Balestriero**[2,3]

[1] Genentech, [2] Brown University, [3] Meta AI, FAIR

## Abstract

Reconstruction and joint-embedding have emerged as two leading paradigms in Self-Supervised Learning (SSL). Reconstruction methods focus on recovering the original sample from a different view in input space. On the other hand, joint-embedding methods align the representations of different views in latent space. Both approaches offer compelling advantages, yet practitioners lack clear guidelines for choosing between them. In this work, we unveil the core mechanisms that distinguish each paradigm. By leveraging closed-form solutions for both approaches, we precisely characterize how the view generation process, *e.g.* data augmentation, impacts the learned representations. We then demonstrate that, unlike supervised learning, both SSL paradigms require a minimal alignment between augmentations and irrelevant features to achieve asymptotic optimality with increasing sample size. Our findings indicate that in scenarios where these irrelevant features have a large magnitude, joint-embedding methods are preferable because they impose a strictly weaker alignment condition compared to reconstruction-based methods. These results not only clarify the trade-offs between the two paradigms but also substantiate the empirical success of joint-embedding approaches on real-world challenging datasets.

## 1 Introduction

Training deep neural networks to extract informative data representations is central to AI. Numerous families of methods pursue this goal [52]. In supervised learning, one does so by prescribing *labels* that encode what is considered informative in the data. While this has been the dominant approach to representation learning over the past decades, it has become clear that labels are often overly specialized. Such specialization prevents learning representations that transfer across an ever-increasing diversity of downstream tasks [24, 37]. Self-Supervised Learning (SSL) has emerged as an alternative that moves away from labels [4, 50, 14]. In SSL, one does not assume *a priori* what is informative; instead, one specifies which variations are uninformative and should be disregarded. Identifying, *a priori*, the invariances a representation should satisfy is a broadly applicable principle. For instance, many downstream tasks involving natural images, such as recognition, counting, and segmentation, are inherently robust to minor changes in color or illumination. Consequently, these tasks benefit from representations that exhibit such invariances, typically encoded through a data-augmentation process. Two primary families of methods have emerged to learn representations using this principle: *reconstruction*-based and *joint-embedding* approaches.

---

[*]Contact: van_assel.hugues@gene.com

39th Conference on Neural Information Processing Systems (NeurIPS 2025).

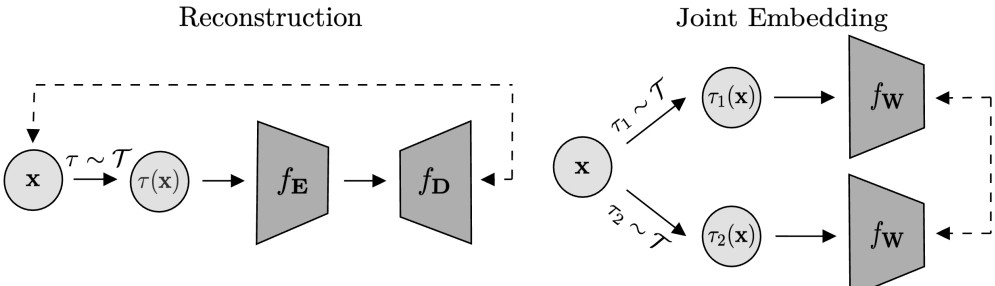

Figure 1: Two self-supervised learning paradigms studied in this work. Left: Reconstruction problem of Equation (SSL-RC): a random augmentation $\tau \sim \mathcal{T}$ is applied to $\mathbf{x}$ to form $\tau(\mathbf{x})$. An encoder $f_{\mathbf{E}}$ together with a decoder $f_{\mathbf{D}}$ is trained to recover $\mathbf{x}$ from $\tau(\mathbf{x})$. Right: Joint embedding problem of Equation (SSL-JE): two independent augmentations $\tau_1, \tau_2 \sim \mathcal{T}$ of the same $\mathbf{x}$ are mapped by $f_{\mathbf{W}}$ to nearby representations, while embeddings of different inputs are pushed apart.

## 1.1 The Reconstruction-based approach

*Reconstruction*-based approaches train models by augmenting an input signal, typically by adding noise or masking, and then training the model to restore the original input [41, 60, 31, 67] (left side of Figure 1). This process encourages the model to learn meaningful internal representations of the data's underlying structure and content to enable successful reconstruction. However, because the learning signal arises from minimizing reconstruction error in the input space, the model is naturally steered toward subspaces that explain the majority of the input's variance [65, 8]. Whether such variance-explaining features are also the most semantically discriminative or useful for downstream tasks depends strongly on the data modality.

In language, reconstruction-based learning, as used in large language models, is highly effective because textual tokens represent compact, semantically meaningful units that already abstract away most low-level variability. Although data quality may vary, language in itself is a highly compressed and rich modality where reconstruction can prove highly successful. Predicting a missing token provides a learning signal that operates directly in semantic space: to succeed, the model must infer the contextual and syntactic relationships that determine meaning, rather than replicate surface patterns. Consequently, minimizing reconstruction error encourages the emergence of abstract relational representations, capturing compositionality, long-range dependencies, and discourse coherence that align closely with human notions of meaning and reasoning [19, 30, 62].

In contrast, in visual domains, variance-explaining features often emphasize aspects of the data that are statistically dominant but semantically shallow. Unlike language, visual data are essentially sensorial recordings of the physical world, capturing raw information without inherent semantic compression. As a result, pixel-level reconstruction objectives tend to drive models toward capturing local statistics and textures that account for most of the input's variance, rather than the higher-order structures and object-level relationships that underpin semantic understanding. This local bias can result in representations that are well-suited for low-level perceptual fidelity but suboptimal for recognition, categorization, or other tasks that depend on global context and semantic abstraction [6, 28]. Consequently, purely reconstruction-based approaches in computer vision often struggle to produce features that generalize well across tasks without additional supervision or adaptation. Fine-tuning is thus frequently necessary to bridge the gap between variance-focused representations and those that encode meaningful, task-relevant semantics [31].

## 1.2 The Joint-embedding approach

*Joint-embedding* methods, in contrast, operate entirely in latent space (right side of Figure 1). Their objective is to produce similar representations for different augmented views of the

same input while ensuring that representations of distinct samples remain dissimilar. This separation can be enforced explicitly through a contrastive loss [14, 33], or implicitly via architectural mechanisms such as self-distillation, stop-gradient operations, momentum encoders, or predictor heads that stabilize training and prevent representational collapse even without negative samples [13, 26, 66, 38].

Unlike *reconstruction*-based approaches, joint-embedding methods do not predict in the input space and are therefore less biased toward capturing high-variance components of the signal. Empirically, joint-embedding frameworks have shown strong performance across domains where the input signal is high-dimensional and semantically diffuse. Successful applications span histopathology [73], Earth observation [61], and video representation learning [9]. Despite this progress, the mechanisms through which latent consistency objectives outperform reconstruction-based ones remain poorly understood, motivating the analysis presented in this work.

### 1.3 Contributions

The critical role of the prediction target in SSL, specifically whether to predict in the input space (*reconstruction*) or the latent representation space (*joint-embedding*), has been demonstrated numerous times [1, 6]. However, it remains unclear when to favor one approach over the other. This work clarifies when to prefer each. Our key findings can be summarized as follows.

1. We derive closed-form solutions for both *reconstruction*-based (Theorem 3.1) and *joint-embedding* (Theorem 3.2) SSL linear models. This enables a precise characterization of data augmentation impacts, analogous to well-known results in supervised learning [11].

2. We then leverage these results to show that optimally aligning the augmentations with the irrelevant components of the input signal can effectively eliminate these components and recover optimal performance for both families of methods (Propositions 4.3 and 4.4). However, in contrast to the supervised learning scenario (Proposition 4.2), simply increasing the sample size cannot overcome any misalignment between the augmentation and the noise (Propositions 4.3 and 4.4).

3. By inspecting the alignment requirements for both *reconstruction* and *joint-embedding* methods, we show that in settings with low-magnitude irrelevant noise features, *reconstruction* methods are preferable, as they require fewer tailored augmentations (Corollary 4.5). Conversely, in scenarios with *high-magnitude* irrelevant noise features, *i.e.* , where such features significantly impact the input signal, *joint-embedding* methods should be preferred, as they impose a strictly weaker alignment condition than reconstruction methods (Corollary 4.5).

4. In Section 5, we experimentally validate these findings on both vectorial and image data. We demonstrate that *joint-embedding* methods such as DINO [13] and BYOL [27] are considerably more robust to severe data corruption than *reconstruction*-based methods like MAE [31] (Section 5.2). In Appendix D, we further provide experimental validation for key results from our theoretical analysis. These experiments show that: (i) SSL methods exhibit significantly greater sensitivity to corruptions compared to supervised learning methods (Appendix D.2 and Figure 2); and (ii) SSL performance in noisy data settings is enhanced by aligning augmentations with the underlying noise (Appendix D.3 and Figure 2).

## 2 Background

Throughout, we consider $n$ samples $\mathbf{X} = (\mathbf{x}_1, \ldots, \mathbf{x}_n)^\top \in \mathbb{R}^{n \times d}$ and associated labels $\mathbf{Y} = (\mathbf{y}_1, \ldots, \mathbf{y}_n)^\top \in \mathbb{R}^{n \times \ell}$. We consider a data augmentation distribution $\mathcal{T}$ defined as a distribution over transformations $\tau : \mathbb{R}^d \to \mathbb{R}^d$.

**Supervised Learning.** For the regression task of predicting labels $\mathbf{Y}$ from observations $\mathbf{X}$, the *augmented empirical risk minimization* problem is as follows:

$$\min_{\mathbf{V}} \frac{1}{n} \sum_{i \in [\![n]\!]} \mathbb{E}_{\tau \sim \mathcal{T}} \left[ \| \mathbf{y}_i - f_{\mathbf{V}}(\tau(\mathbf{x}_i)) \|_2^2 \right] . \tag{SL}$$

Interestingly, when using a linear model $f_{\mathbf{V}} : \mathbf{x} \mapsto \mathbf{V}\mathbf{x}$ with $\mathbf{V} \in \mathbb{R}^{\ell \times d}$, the effect of data augmentation in Equation (SL) can be explicitly characterized as a Tikhonov regularization problem as shown in Lemma B.1 [7, 46, 11] which proof is provided in Appendix B:

$$\frac{1}{n} \sum_{i \in [\![n]\!]} \mathbb{E}_{\tau \sim \mathcal{T}} \left[ \|\mathbf{y}_i - \mathbf{V}\tau(\mathbf{x}_i)\|_2^2 \right] = \|\mathbf{V}\|_{\boldsymbol{\Sigma}}^2 + \frac{1}{n} \sum_{i \in [\![n]\!]} \|\mathbf{y}_i - \mathbf{V}\mathbb{E}_{\tau \sim \mathcal{T}} \left[ \tau(\mathbf{x}_i) \right] \|_2^2 \qquad (1)$$

where $\|\mathbf{V}\|_{\boldsymbol{\Sigma}}^2 = \mathrm{Tr}(\mathbf{V}\boldsymbol{\Sigma}\mathbf{V}^\top)$ and

$$\boldsymbol{\Sigma} := \frac{1}{n} \sum_i \mathbb{E}_{\tau \sim \mathcal{T}} \left[ \tau(\mathbf{x}_i)\tau(\mathbf{x}_i)^\top \right] - \mathbb{E}_{\tau \sim \mathcal{T}} \left[ \tau(\mathbf{x}_i) \right] \mathbb{E}_{\tau \sim \mathcal{T}} \left[ \tau(\mathbf{x}_i) \right]^\top \qquad \text{(Cov)}$$

denotes the covariance of the augmented samples. Therefore, the effect of data augmentation within supervised learning using linear models is well understood from a theoretical standpoint.

**Lack of Foundations in Self-Supervised Learning.** Similar results are lacking for SSL, where the explicit effect of data augmentation for linear models has not been rigorously studied. Despite recent efforts to elucidate the underlying principles [64, 56, 39, 36, 29, 22, 71, 63], these methods remain only superficially understood [53]. A robust statistical framework is still lacking to fully comprehend SSL methods and to position them relative to their supervised learning counterparts [3]. Key open questions involve precisely characterizing the role of data augmentation in shaping final representations within both *reconstruction* and *joint-embedding* frameworks. This work aims to lay the foundation for filling this gap.

## 3 Augmentation-Aware Closed-Form Solutions

In this section, we derive closed-form solutions for the two main families of SSL methods: *reconstruction-based* and *joint-embedding* approaches. To the best of our knowledge, the following results are the first instances of closed-form solutions for SSL that are directly parameterized by the data augmentation structure. This stands in contrast to previous solutions highlighted in [5], which focused on the dependency graph between augmented samples and were unaware of augmentations. These results will then allow us to analyze how augmentations affect the learned representations in both families of SSL methods.

In line with previous theoretical works focused on analytical tractability [12, 5, 47, 54, 57], we study models that are linear in their parameters. Note that these models can produce arbitrary nonlinear predictions via appropriate feature maps [3] and are known to describe regimes of wide neural networks [44].

### 3.1 Reconstruction-Based Self-Supervised Learning

We first consider *reconstruction*-based SSL models. The problem can be framed as follows, where $\mathcal{T}$ is the data augmentation distribution:

$$\min_{\mathbf{E},\mathbf{D}} \quad \frac{1}{n} \sum_{i \in [\![n]\!]} \mathbb{E}_{\tau \sim \mathcal{T}} \left[ \|\mathbf{x}_i - f_{\mathbf{D}}(f_{\mathbf{E}}(\tau(\mathbf{x}_i)))\|_2^2 \right] . \qquad \text{(SSL-RC)}$$

In this formulation, each data sample is augmented and then passes through an encoder $f_{\mathbf{E}}$, followed by a decoder $f_{\mathbf{D}}$. The objective is to minimize the reconstruction error between the original sample and the reconstructed one. This methodology is analogous to the *Denoising Auto-Encoder* [59], *Masked Auto-Encoder* [32] and similar frameworks. Interestingly, the *reconstruction* problem can be solved in closed form when considering linear models for both encoder and decoder. All proofs can be found in Appendix A.

**Theorem 3.1.** *Let* $\overline{\mathbf{x}}_i := \mathbb{E}_{\tau \sim \mathcal{T}} \left[ \tau(\mathbf{x}_i) \right]$ *for each* $i \in [\![n]\!]$, *and define* $\overline{\mathbf{X}} := (\overline{\mathbf{x}}_1, \ldots, \overline{\mathbf{x}}_n)^\top$. *Assume that* $\frac{1}{n}\overline{\mathbf{X}}^\top\overline{\mathbf{X}} + \boldsymbol{\Sigma}$ *is positive definite where* $\boldsymbol{\Sigma}$ *is defined in Equation* (Cov). *Consider the singular value decomposition:*

$$\frac{1}{n}\mathbf{X}^\top\overline{\mathbf{X}} \left( \frac{1}{n}\overline{\mathbf{X}}^\top\overline{\mathbf{X}} + \boldsymbol{\Sigma} \right)^{-\frac{1}{2}} = \mathbf{R}\boldsymbol{\Phi}\mathbf{P}^\top \qquad (2)$$

where $\mathbf{R} \in \mathbb{R}^{d \times d}$ and $\mathbf{P} \in \mathbb{R}^{d \times d}$ are orthogonal and $\mathbf{\Phi} := \operatorname{diag}(\phi_1, \ldots, \phi_d)$ with $\phi_1 \geq \cdots \geq \phi_d \geq 0$. Solutions of Equation (SSL-RC) for $f_{\mathbf{E}} : \mathbf{x} \mapsto \mathbf{Ex}$ and $f_{\mathbf{D}} : \mathbf{x} \mapsto \mathbf{Dx}$ take the form:

$$\mathbf{E}^\star = \mathbf{T}\mathbf{P}_k^\top \left( \tfrac{1}{n}\overline{\mathbf{X}}^\top \overline{\mathbf{X}} + \mathbf{\Sigma} \right)^{-\frac{1}{2}} \quad and \quad \mathbf{D}^\star = \mathbf{R}_k \mathbf{\Phi}_k \mathbf{T}^{-1}, \tag{3}$$

where $\mathbf{T}$ is any invertible matrix in $\mathbb{R}^{k \times k}$, $\mathbf{P}_k$ and $\mathbf{R}_k$ are the first $k$ columns of $\mathbf{P}$ and $\mathbf{R}$ respectively, and $\mathbf{\Phi}_k = \operatorname{diag}(\phi_1, \ldots, \phi_k)$.

## 3.2 Joint-Embedding-Based Self-Supervised Learning

We now consider a *joint-embedding* SSL problem formulated as follows, where $f_{\mathbf{W}}$ is the SSL model and $\mathcal{T}$ is the data augmentation distribution:

$$\min_{\mathbf{W}} \quad \frac{1}{n} \sum_{i \in [\![n]\!]} \mathbb{E}_{\tau_1, \tau_2 \sim \mathcal{T}} \left[ \| f_{\mathbf{W}}(\tau_1(\mathbf{x}_i)) - f_{\mathbf{W}}(\tau_2(\mathbf{x}_i)) \|_2^2 \right],$$

$$\text{subject to} \quad \frac{1}{n} \sum_{i \in [\![n]\!]} \mathbb{E}_{\tau \sim \mathcal{T}} \left[ f_{\mathbf{W}}(\tau(\mathbf{x}_i)) f_{\mathbf{W}}(\tau(\mathbf{x}_i))^\top \right] = \mathbf{I}_k. \tag{SSL-JE}$$

In the above Equation (SSL-JE), the objective represents the usual invariance term which ensures consistency between two augmented views of the same sample and is a common component of *joint-embedding* methods. The constraint enforces orthonormality in the learned representations, promoting diversity in the representation space [64] and thus preventing collapse. Most *joint-embedding* models incorporate a similar repulsion term, either explicitly within the loss function, such as in Barlow-Twins [70], SimCLR [14], VICReg [10], and MoCo [33], or implicitly through architectural design choices, as demonstrated by BYOL [26] and DINO [13]. In our case, we rely on the sum of the outer products of the representation vectors. This approach closely resembles VICReg [10], specifically its covariance regularization term. Interestingly, under the unifying formalism presented in [25], most popular *joint-embedding* methods can be framed with this simple repulsive term.

The problem of Equation (SSL-JE) can also be solved in closed form when considering a linear SSL model, as formalized below.

**Theorem 3.2.** *Let* $\mathbf{S} := \frac{1}{n} \sum_i \mathbb{E}_{\tau \sim \mathcal{T}} \left[ \tau(\mathbf{x}_i)\tau(\mathbf{x}_i)^\top \right]$, $\mathbf{G} := \frac{1}{n} \sum_i \mathbb{E}_{\tau \sim \mathcal{T}} \left[ \tau(\mathbf{x}_i) \right] \mathbb{E}_{\tau \sim \mathcal{T}} \left[ \tau(\mathbf{x}_i) \right]^\top$. *Assume that* $\mathbf{S}$ *is positive definite. Consider the eigendecomposition:*

$$\mathbf{S}^{-\frac{1}{2}} \mathbf{G} \mathbf{S}^{-\frac{1}{2}} = \mathbf{Q} \mathbf{\Omega} \mathbf{Q}^\top \tag{4}$$

*where* $\mathbf{\Omega} = \operatorname{diag}(\omega_1, \ldots, \omega_d)$ *with* $\omega_1 \geq \cdots \geq \omega_d$. *Solutions of Equation* (SSL-JE) *for a linear model* $f_{\mathbf{W}} : \mathbf{x} \mapsto \mathbf{Wx}$ *take the form:*

$$\mathbf{W}^\star = \mathbf{U} \mathbf{Q}_k^\top \mathbf{S}^{-\frac{1}{2}}, \tag{5}$$

*where* $\mathbf{Q}_k = (\mathbf{q}_1, \ldots, \mathbf{q}_k)$ *and* $\mathbf{U}$ *is any orthogonal matrix of size* $k \times k$.

## 4  Augmentation Alignment Requirements in Self-Supervised Learning

In this section, we build on the results of Section 3 to evaluate the ability of both families of SSL models to reach optimal performance. To define such notion of optimality, we model our data as having $k$ *important signal components* and $d - k$ *pure noise components*. These *noise components* represent the variations that SSL methods are typically designed to be invariant to, *e.g.* background noise for image classification tasks. An ideal SSL encoder would retain the $k$ *informative important dimensions* while discarding the $d - k$ *noise components*.

We formalize this scenario in Section 4.1, where a parameter $\alpha$ is introduced to control the alignment between the irrelevant features and the augmentations. In Section 4.2, we demonstrate that supervised learning models can effectively achieve optimal performance either when augmentations are well aligned with the irrelevant noise features at finite sample

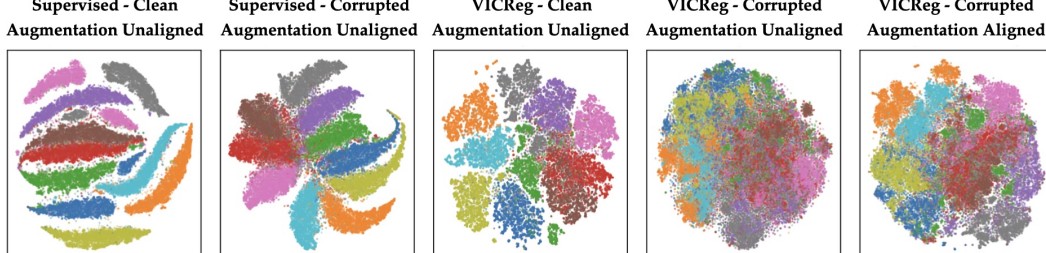

| Supervised - Clean Augmentation Unaligned | Supervised - Corrupted Augmentation Unaligned | VICReg - Clean Augmentation Unaligned | VICReg - Corrupted Augmentation Unaligned | VICReg - Corrupted Augmentation Aligned |

Figure 2: Injecting corruption-aligned noise into data augmentation improves SSL representation quality on corrupted CIFAR-10. Thus **aligning augmentations with the irrelevant components in the data is crucial in SSL**. t-SNE [58] visualizations of (left to right): (1) Supervised features (penultimate layer), clean data. (2) Supervised features (penultimate layer), fog-corrupted (severity 5). (3) VICReg representations, clean data. (4) VICReg representations, fog-corrupted (severity 5). (5) VICReg representations, fog-corrupted (severity 5) with fog noise (severity 1) injection during augmentation. Unlike supervised features, VICReg representations degrade significantly under corruption (compare 3 and 4). Injecting noise in the data augmentation (5) enhances class separability.

sizes or when the sample size is large, regardless of the augmentation employed. In Section 4.3, we show that, unlike supervised learning, SSL necessitates a sufficiently good alignment to achieve optimal performance, even in the infinite sample limit. Finally in Section 4.4, we compare the alignment requirements of *joint-embedding* and *reconstruction*-based SSL methods, thus providing insights into the characteristics of both families of methods.

## 4.1 Data, Noise and Augmentation

We consider an input dataset with two parts: *important features* and *irrelevant noise*. Optimal performance on downstream tasks is achieved when using only the *important features*. Let $\mathbf{X} = \mathbf{L}\mathbf{K}\mathbf{Q}^\top$ be the singular value decomposition of the *important features*, where $\mathbf{K} = \mathrm{diag}(\kappa_1, \ldots, \kappa_d)$ is the diagonal matrix of singular values. Each sample $\mathbf{x}_i$ is corrupted by additive Gaussian noise constituting the *irrelevant* features:

$$\forall\, i \in [\![n]\!], \quad \widetilde{\mathbf{x}}_i = \mathbf{x}_i + \boldsymbol{\gamma}_i, \quad \boldsymbol{\gamma}_i \sim \mathcal{N}(\mathbf{0}, \boldsymbol{\Gamma}), \tag{6}$$

with $\boldsymbol{\gamma}_i$ drawn independently across $i \in [\![n]\!]$ and where $\boldsymbol{\Gamma} \in \mathbb{R}^{d \times d}$ is positive semi-definite. For simplicity, we assume that $\boldsymbol{\Gamma}$ is diagonalized by the same orthonormal matrix $\mathbf{Q}$ from the SVD above *i.e.* $\boldsymbol{\Gamma} = \mathbf{Q}\boldsymbol{\Lambda_\Gamma}\mathbf{Q}^\top$ where $\boldsymbol{\Lambda_\Gamma} = \mathrm{diag}(\lambda_1^{\boldsymbol{\Gamma}}, \ldots, \lambda_d^{\boldsymbol{\Gamma}})$. The matrix $\widetilde{\mathbf{X}} = (\widetilde{\mathbf{x}}_1, \ldots, \widetilde{\mathbf{x}}_n)^\top \in \mathbb{R}^{n \times d}$ then forms the *corrupted* input data. We assume that the *important features* are concentrated in exactly $k \geq 1$ components, meaning $\kappa_i > 0$ for $i \in [\![k]\!]$ and $\kappa_i = 0$ for $i > k$. These components are referred to as the *important components*. Additionally, we assume that the *irrelevant noise* are null in these $k$ components, i.e. $\lambda_i^{\boldsymbol{\Gamma}} = 0$ for all $i \in [\![k]\!]$, and strictly positive otherwise, i.e. $\lambda_i^{\boldsymbol{\Gamma}} > 0$ for $i \in [\![k+1 : d]\!]$. We refer to the $[\![k+1 : d]\!]$ components as the *noise components*.

**Data augmentation.** Let $\boldsymbol{\Theta} \in \mathbb{R}^{d \times d}$ be positive semi-definite and diagonalized by $\mathbf{Q}$ *i.e.* $\boldsymbol{\Theta} = \mathbf{Q}\boldsymbol{\Lambda_\Theta}\mathbf{Q}^\top$ where $\boldsymbol{\Lambda_\Theta} = \mathrm{diag}(\lambda_1^{\boldsymbol{\Theta}}, \ldots, \lambda_d^{\boldsymbol{\Theta}})$. We consider the augmentation distribution,

$$\forall\, \alpha \geq 0, \ \ \mathcal{T}(\alpha) := \left\{ \tau : \mathbb{R}^d \to \mathbb{R}^d \,\middle|\, \tau(\mathbf{x}) = \mathbf{x} + \boldsymbol{\theta} + \alpha\,\boldsymbol{\gamma}, \ \boldsymbol{\theta} \sim \mathcal{N}(\mathbf{0}, \boldsymbol{\Theta}), \ \boldsymbol{\gamma} \sim \mathcal{N}(\mathbf{0}, \boldsymbol{\Gamma}) \right\}, \tag{7}$$

where $\boldsymbol{\theta}$ and $\boldsymbol{\gamma}$ are drawn independently for each transformation. Note that the term $\boldsymbol{\gamma}$ follows the same distribution as the noisy irrelevant features. Increasing the magnitude of $\alpha$ thus aligns the data augmentation with these irrelevant features.

*Remark* 4.1. One can extend our result to augmentations beyond Gaussians by considering any augmentation of *covariance* (defined in Cov) $\boldsymbol{\Theta} + \alpha^2 \boldsymbol{\Gamma}$ [46].

## 4.2 Supervised Learning Consistency Regardless of Augmentations

We first analyze the behavior of supervised learning models by identifying regimes in which the supervised model effectively disregards the *noisy irrelevant features in* $\widetilde{\mathbf{X}}$. This provides the foundation for pinpointing key differences between supervised and SSL models in Section 4.3. We rely on the data augmentation $\mathcal{T}(\alpha)$, where $\alpha \in \mathbb{R}_+$ controls the alignment between the data corruption and the augmentation process as presented in Section 4.1.

**Proposition 4.2.** *[Supervised Learning] Let $\mathbf{V}^\star$ (resp. $\widetilde{\mathbf{V}}^\star$) be the linear model solving Equation* (SL) *with augmentation $\mathcal{T}(\alpha)$ for $\mathbf{X}$ (resp. the corrupted $\widetilde{\mathbf{X}}$). The limit:*

$$\widetilde{\mathbf{V}}^\star \xrightarrow[a.s.]{} \mathbf{V}^\star \tag{8}$$

*holds almost surely in either of the following regimes:*

- *as $\alpha \to +\infty$ (perfect augmentation-noise alignment) for any fixed sample size $n \in \mathbb{N}$.*

- *as $n \to +\infty$ (infinite samples) for any fixed alignment $\alpha \geq 0$.*

The above result shows that, when performing supervised learning with *corrupted* data, the model can achieve the same performance as if it were trained only on the *important features* (thus achieving optimal performance) if either of the following conditions holds: i) The data augmentation process is well aligned with the noise corrupting the inputs ($\alpha$ large). ii) A sufficiently large sample size is available to compensate for any misalignment between the augmentation and the input noise ($n$ large).

## 4.3 Self-Supervised Learning Requires Aligned Augmentation and Noise

Building on the closed-form expressions for SSL provided in Theorems 3.1 and 3.2, we are now interested in studying the ability of SSL models to achieve optimal performance when trained on the *corrupted* dataset $\widetilde{\mathbf{X}}$, as defined in Section 4.1.

**Proposition 4.3.** *[Reconstruction] Let $\mathbf{E}^\star$ (resp. $\widetilde{\mathbf{E}}^\star$) be the linear (encoder) model solving Equation* (SSL-RC) *for $\mathbf{X}$ (resp. the corrupted $\widetilde{\mathbf{X}}$). The limit:*

$$\widetilde{\mathbf{E}}^\star \xrightarrow[a.s.]{} \mathbf{E}^\star \tag{9}$$

*holds[2] almost surely in either of the following regimes:*

- *as $\alpha \to +\infty$ (perfect augmentation-noise alignment) for any fixed sample size $n \in \mathbb{N}$.*

- *as $n \to +\infty$ (infinite samples), if and only if the alignment $\alpha \geq 0$ satisfies:*

$$\alpha^2 > \alpha_{\mathrm{RC}}^2 := \max_{i \in [\![k+1:d]\!]} \frac{\lambda_i^{\mathbf{\Gamma}}}{\eta^2} - \frac{\lambda_i^{\mathbf{\Theta}}}{\lambda_i^{\mathbf{\Gamma}}} - 1 \quad where \quad \eta = \min_{i \in [\![k]\!]} \frac{\frac{1}{n}\kappa_i^2}{\sqrt{\frac{1}{n}\kappa_i^2 + \lambda_i^{\mathbf{\Theta}}}} \,. \tag{10}$$

**Proposition 4.4.** *[Joint-Embedding] Let $\mathbf{W}^\star$ (resp. $\widetilde{\mathbf{W}}^\star$) be the linear model solving Equation* (SSL-JE) *for $\mathbf{X}$ (resp. the corrupted $\widetilde{\mathbf{X}}$). The limit:*

$$\widetilde{\mathbf{W}}^\star \xrightarrow[a.s.]{} \mathbf{W}^\star \tag{11}$$

*holds[3] almost surely in either of the following regimes:*

- *as $\alpha \to +\infty$ (perfect augmentation-noise alignment) for any fixed sample size $n \in \mathbb{N}$.*

---

[2]Up to an arbitrary invertible matrix (i.e., if $\mathbf{E}^\star$ is a solution, so is $\mathbf{T}\mathbf{E}^\star$ for any $k \times k$ invertible matrix $\mathbf{T}$).

[3]Up to an arbitrary orthogonal rotation (i.e., if $\mathbf{W}^\star$ is a solution, so is $\mathbf{U}\mathbf{W}^\star$ for any $k \times k$ orthogonal matrix $\mathbf{U}$).

- *as $n \to +\infty$ (infinite samples), if and only if the alignment $\alpha \geq 0$ satisfies:*

$$\alpha^2 > \alpha^2_{\mathrm{JE}} := \max_{i \in [\![k+1:d]\!]} \frac{1-\delta}{\delta} - \frac{\lambda_i^{\Theta}}{\lambda_i^{\Gamma}} \quad where \quad \delta = \min_{i \in [\![k]\!]} \frac{\frac{1}{n}\kappa_i^2}{\frac{1}{n}\kappa_i^2 + \lambda_i^{\Theta}} . \tag{12}$$

The above Propositions 4.3 and 4.4 reveal that, when augmentations are well aligned with the noise *i.e.* when $\alpha$ is large enough, undesired *noise components* are removed from the obtained SSL representation for both families of models, even when combined with other augmentations. However, the alignment requirement persists even as the sample size $n$ becomes arbitrarily large. Therefore, in SSL, achieving optimal performance requires that the data augmentation process be sufficiently well aligned with the *irrelevant noise features* in the data. This marks a key difference from supervised models. Unlike SSL, supervised models can overcome misalignment between augmentations and noise with enough samples, as it learns robustness by observing different noise realizations across identically labeled data. This underscores the critical role of augmentations in SSL. Figure 2 illustrates this by visualizing the embedding spaces of a supervised model and the VICReg [10] SSL model, revealing the latter's susceptibility to noise when augmentations lack proper alignment. This finding is consistent with an empirical study by [51], which concluded that improving augmentations is more impactful than altering architectural designs.

Interestingly, the above results reveal that *reconstruction* (Proposition 4.3) and *joint-embedding* (Proposition 4.4) methods exhibit different alignment requirements to achieve optimal performance. We next analyze these differences.

## 4.4 Comparison of Joint-Embedding and Reconstruction-Based Methods via Augmentation Alignment Requirement

Leveraging Propositions 4.3 and 4.4, we obtain the following key result.

**Corollary 4.5.** *Let $\alpha_{\mathrm{JE}}$, $\delta$, $\alpha_{\mathrm{RC}}$, and $\eta$ be defined as in Proposition 4.4 and Proposition 4.3.*

- *If $\max_{i \in [\![k+1:d]\!]} \lambda_i^{\Gamma} < \dfrac{\eta^2}{\delta}$ (low noise), then $\alpha_{\mathrm{JE}} > \alpha_{\mathrm{RC}}$ (reconstruction is preferable).*

- *If $\min_{i \in [\![k+1:d]\!]} \lambda_i^{\Gamma} > \dfrac{\eta^2}{\delta}$ (high noise), then $\alpha_{\mathrm{JE}} < \alpha_{\mathrm{RC}}$ (joint-embedding is preferable).*

This result shows that when the spectral norm of the noise covariance $\mathbf{\Gamma}$ is small, *reconstruction*-based methods impose a less stringent alignment compared to *joint-embedding* methods. Conversely, when the noise magnitude is large, *joint-embedding* methods exhibit lower sensitivity to the augmentation-noise alignment compared to their *reconstruction*-based counterparts. Ultimately, the goal is to minimize the alignment requirement, as the *irrelevant components* are typically unknown in real-world applications.

*Reconstruction*-based approaches are therefore well-suited for scenarios with weak, *irrelevant noise features*. Intuitively, the core *important components* in such settings possess the greatest magnitude and are consequently prioritized by the model during reconstruction. Due to their reconstruction objective, these methods depend less on data augmentations, which explains their superior performance and robustness over *joint-embedding* techniques in this particular context.

However, when strong *noise features* obscure the *important features* within the raw input signal, *joint-embedding* methods demonstrate greater robustness. This is because *joint-embedding* techniques prioritize latent space prediction, thereby bypassing the need to reconstruct *irrelevant noise components* as model outputs. Consequently, in real-world scenarios where the extent of *irrelevant features* (typically image backgrounds or experimental batch effects in scRNA-seq data) cannot be precisely quantified, *joint-embedding* approaches appear more reliable. This preference is further underscored by the principle that high-amplitude noise naturally exerts a more significant impact on the final representation than low-amplitude noise, making robustness crucial when noise levels are uncertain. This practical advantage also explains the community's preference for *joint-embedding* approaches in challenging datasets [73, 43, 20].

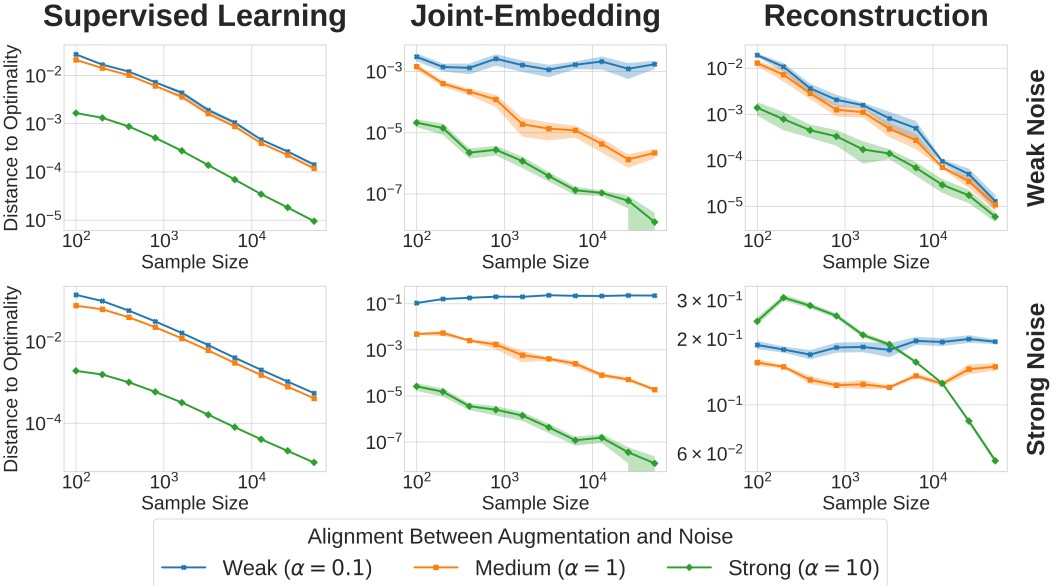

Figure 3: Performance of linear supervised and SSL models (Sections 3.1 and 3.2 and Theorems 3.1 and 3.2) on MNIST corrupted with synthetic Gaussian noise (Section 4.1) with various augmentation alignment $\alpha$ Section 4. Each subplot's y-axis is the absolute difference of supervised linear probing loss (on clean vs. corrupted data) and its x-axis is the sample size $n$. This figure highlights that *joint-embedding* is preferable to *reconstruction* in the presence of strong *irrelevant noise features*. Conversely, *reconstruction* requires less tailored augmentation when dealing with weak *irrelevant noise features*. Weak noise corresponds to $\lambda^{\mathbf{\Gamma}}_{\max} = 10^3$ and strong noise to $\lambda^{\mathbf{\Gamma}}_{\max} = 10^6$ (details in Appendix C).

> **Key takeaway.** When *irrelevant features* have low magnitude and there is limited prior information on effective augmentations, *reconstruction* is preferable. In contrast, when these *irrelevant features* are non-negligible (as is common with real-world data) or effective augmentations can be identified, *joint-embedding* is preferable.

## 5   Experiments

This section validates the theoretical findings of Section 4 through experiments on linear models (Section 5.1) and deep networks (Section 5.2). These experiments confirm that the results from the linear model are consistent with those observed in the nonlinear setting of deep networks.

### 5.1   Experiments With Linear Models

We first validate the theoretical results of Section 4 through experiments with linear models. Data features are corrupted by adding synthetic Gaussian noise, allowing us to precisely control the noise magnitude and its alignment with data augmentations. The details of our experimental design are provided in Appendix C. Our primary results are illustrated in Figure 3. This figure effectively illustrates contrasting behaviors between the various types of methods as sample size and noise magnitude vary. On the left, one can notice that the supervised model achieves optimal performance with either increasing sample size or increasing alignment, with any augmentation and regardless of the *noise magnitude*, confirming the result of Proposition 4.2. In contrast, SSL models exhibit different sensitivities, as predicted in Propositions 4.3 and 4.4. The middle panel shows that *joint-embedding* indeed requires a minimal alignment between augmentation and noise to reach optimal performance (as predicted in Proposition 4.4). Notably, *joint-embedding* maintains robustness even with increasing noise magnitude, as shown in Corollary 4.5. On the right panel, we can see

Table 1: Linear probing top1 accuracy scores of MAE, DINO, and SimCLR on ImageNet with various corruptions [35] and relative performance drop from severity 1 to 5.

| Method | Pixelate Corruption | | | | Gaussian Noise Corruption | | | | Zoomblur Corruption | | | |
|---|---|---|---|---|---|---|---|---|---|---|---|---|
| | Sev. 1 | Sev. 3 | Sev. 5 | Drop (%) | Sev. 1 | Sev. 3 | Sev. 5 | Drop (%) | Sev. 1 | Sev. 3 | Sev. 5 | Drop (%) |
| BYOL | 66.7 | 61.3 | 58.7 | 12.0 | 67.2 | 63.1 | 56.4 | 16.1 | 70.1 | 67.0 | 63.8 | 9.0 |
| DINO | 68.7 | 64.9 | 60.2 | 12.4 | 67.6 | 62.4 | 59.0 | 12.7 | 69.4 | 67.2 | 64.9 | 6.5 |
| MAE | 64.9 | 52.3 | 46.8 | 27.9 | 61.6 | 46.7 | 44.8 | 27.3 | 64.1 | 58.4 | 51.3 | 20.0 |

that under weak noise conditions, *reconstruction* is robust to the choice of augmentation. However, as noise becomes stronger, *reconstruction* performance degrades and necessitates a strong alignment between augmentation and noise. These observations confirm the results of Propositions 4.3 and 4.4 and Corollary 4.5. These findings are further supported by experiments on other datasets, including Fashion-MNIST, Kuzushiji-MNIST and the single-cell RNA-seq data from [49], presented in Figures 4 to 7 in Appendix C. All experimental outcomes align with and confirm the theoretical insights detailed in Section 4.

## 5.2 Experiments with Deep Networks on Images with Various Corruptions

We conduct experiments using both ViT [21] and ResNet [34] architectures. Our evaluation focuses on top-1 linear probing accuracy on ImageNet-100 [18]. ImageNet images inherently contain non-negligible features irrelevant to the classification task [6] (*e.g.* background noise elements), which contributes to the superior performance of *joint-embedding* methods over *reconstruction* methods, as demonstrated in several benchmarks [17, 15]. To further introduce and control the magnitude of *irrelevant noise features*, we utilize the ImageNet-C corruptions [35], which offers corruptions at various severity levels. The results are presented in Table 1. The performance of MAE (ViT) [31], which uses a *reconstruction* objective, is much more affected by the increasing corruptions than DINO (ViT) [13] and BYOL (ResNet) [26], which use a *joint-embedding* objective. Indeed, there is a 25.1% average drop in accuracy for MAE when the severity of the corruption increases from 1 to 5, while DINO and BYOL only experience a 10.5% drop and 12.4% drop, respectively. All experimental details are provided in Appendix D.1.

We perform further ablation studies in the appendix (Appendix D) highlighting this paper's key results. Specifically, our experiments confirm that: (i) SSL is considerably more sensitive to the alignment between augmentations and noise than supervised learning (see Appendix D.2), and (ii) aligning augmentations with the underlying noise can enhance the performance of SSL models in noisy data settings (see Appendix D.3).

## 6 Conclusion

A growing body of work demonstrates that joint-embedding methods often outperform reconstruction methods on real-world datasets, particularly where extracting useful features for downstream tasks from raw signals is challenging [40, 2]. This is further supported by a consistent empirical finding: reconstruction methods typically necessitate fine-tuning to address the inherent misalignment between the features they learn and those that are perceptually useful for downstream applications [72, 6, 45, 68]. In this work, we have established a theoretical framework to explain this phenomenon. Our analysis provides clear guidelines for practitioners: opt for *reconstruction* when *irrelevant* components show low variance and prior information on effective augmentations is scarce. In contrast, prefer *joint-embedding* when these *irrelevant* components have high variance magnitude, as is common with real-world data, or when effective augmentations are either readily available or can be found through cross-validation.

This paper offers a basis for future work. One could consider extending these results to finite sample size settings to precisely characterize the interplay between sample complexity and augmentation in SSL.

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

## A    Proofs

### A.1    Proof of Theorem 3.1

**Theorem 3.1.** *Let* $\overline{\mathbf{x}}_i \coloneqq \mathbb{E}_{\tau \sim \mathcal{T}}\big[\tau(\mathbf{x}_i)\big]$ *for each* $i \in [\![n]\!]$, *and define* $\overline{\mathbf{X}} \coloneqq (\overline{\mathbf{x}}_1, \dots, \overline{\mathbf{x}}_n)^\top$. *Assume that* $\frac{1}{n}\overline{\mathbf{X}}^\top \overline{\mathbf{X}} + \mathbf{\Sigma}$ *is positive definite where* $\mathbf{\Sigma}$ *is defined in Equation* (Cov). *Consider the singular value decomposition:*

$$\tfrac{1}{n}\mathbf{X}^\top\overline{\mathbf{X}}\left(\tfrac{1}{n}\overline{\mathbf{X}}^\top\overline{\mathbf{X}} + \mathbf{\Sigma}\right)^{-\frac{1}{2}} = \mathbf{R}\mathbf{\Phi}\mathbf{P}^\top \tag{2}$$

*where* $\mathbf{R} \in \mathbb{R}^{d \times d}$ *and* $\mathbf{P} \in \mathbb{R}^{d \times d}$ *are orthogonal and* $\mathbf{\Phi} \coloneqq \mathrm{diag}(\phi_1, \dots, \phi_d)$ *with* $\phi_1 \geq \cdots \geq \phi_d \geq 0$. *Solutions of Equation* (SSL-RC) *for* $f_{\mathbf{E}} : \mathbf{x} \mapsto \mathbf{E}\mathbf{x}$ *and* $f_{\mathbf{D}} : \mathbf{x} \mapsto \mathbf{D}\mathbf{x}$ *take the form:*

$$\mathbf{E}^\star = \mathbf{T}\mathbf{P}_k^\top\left(\tfrac{1}{n}\overline{\mathbf{X}}^\top\overline{\mathbf{X}} + \mathbf{\Sigma}\right)^{-\frac{1}{2}} \quad and \quad \mathbf{D}^\star = \mathbf{R}_k\mathbf{\Phi}_k\mathbf{T}^{-1}, \tag{3}$$

*where* $\mathbf{T}$ *is any invertible matrix in* $\mathbb{R}^{k \times k}$, $\mathbf{P}_k$ *and* $\mathbf{R}_k$ *are the first* $k$ *columns of* $\mathbf{P}$ *and* $\mathbf{R}$ *respectively, and* $\mathbf{\Phi}_k = \mathrm{diag}(\phi_1, \dots, \phi_k)$.

*Proof.* Relying on the result of Lemma B.1, we can rewrite the objective as:

$$\tfrac{1}{n}\sum_{i\in[\![n]\!]}\mathbb{E}_{\tau\sim\mathcal{T}}\left[\|\mathbf{x}_i - \mathbf{D}\mathbf{E}\tau(\mathbf{x}_i)\|_2^2\right] = \tfrac{1}{n}\sum_{i\in[\![n]\!]}\|\mathbf{x}_i - \mathbf{D}\mathbf{E}\mathbb{E}_{\tau\sim\mathcal{T}}[\tau(\mathbf{x}_i)]\|_2^2 + \|\mathbf{D}\mathbf{E}\|_{\mathbf{\Sigma}}^2, \tag{13}$$

where $\mathbf{\Sigma} = \frac{1}{n}\sum_i \mathbb{E}_{\tau\sim\mathcal{T}}\left[\tau(\mathbf{x}_i)\tau(\mathbf{x}_i)^\top\right] - \mathbb{E}_{\tau\sim\mathcal{T}}\left[\tau(\mathbf{x}_i)\right]\mathbb{E}_{\tau\sim\mathcal{T}}\left[\tau(\mathbf{x}_i)\right]^\top$. Define per-sample means $\overline{\mathbf{x}}_i \coloneqq \mathbb{E}_{\tau\sim\mathcal{T}}[\tau(\mathbf{x}_i)]$ and stack them into $\overline{\mathbf{X}} \coloneqq (\overline{\mathbf{x}}_1, \dots, \overline{\mathbf{x}}_n)^\top$. The objective can be rewritten as:

$$\min_{\mathbf{E}\in\mathbb{R}^{k\times d}, \mathbf{D}\in\mathbb{R}^{d\times k}} \quad \tfrac{1}{n}\|\mathbf{X} - \overline{\mathbf{X}}\mathbf{E}^\top\mathbf{D}^\top\|_F^2 + \|\mathbf{D}\mathbf{E}\|_{\mathbf{\Sigma}}^2. \tag{14}$$

We consider the equivalent problem on $\mathbf{M} = \mathbf{D}\mathbf{E} \in \mathbb{R}^{d\times d}$:

$$\min_{\mathbf{M}\in\mathbb{R}^{d\times d}} \quad \tfrac{1}{n}\|\mathbf{X} - \overline{\mathbf{X}}\mathbf{M}^\top\|_F^2 + \|\mathbf{M}\|_{\mathbf{\Sigma}}^2 \quad \text{s.t.} \quad \mathrm{rank}(\mathbf{M}) \leq k. \tag{15}$$

In the above, the rank constraint captures the fact that $\mathbf{M}$ has the form $\mathbf{M} = \mathbf{D}\mathbf{E}$ with $\mathbf{D} \in \mathbb{R}^{d\times k}$ and $\mathbf{E} \in \mathbb{R}^{k\times d}$. The objective can be developed as

$$\tfrac{1}{n}\mathrm{Tr}(\mathbf{X}^\top\mathbf{X}) + \tfrac{1}{n}\mathrm{Tr}(\mathbf{M}\overline{\mathbf{X}}^\top\overline{\mathbf{X}}\mathbf{M}^\top) - \tfrac{2}{n}\mathrm{Tr}(\mathbf{M}\overline{\mathbf{X}}^\top\mathbf{X}) + \mathrm{Tr}(\mathbf{M}\mathbf{\Sigma}\mathbf{M}^\top). \tag{16}$$

Keeping only the terms that depend on $\mathbf{M}$, we can rewrite the problem as

$$\min_{\mathbf{M}\in\mathbb{R}^{d\times d}} \quad \mathrm{Tr}\left(\mathbf{M}\left(\tfrac{1}{n}\overline{\mathbf{X}}^\top\overline{\mathbf{X}}+\mathbf{\Sigma}\right)\mathbf{M}^\top\right) - \tfrac{2}{n}\mathrm{Tr}(\mathbf{M}\overline{\mathbf{X}}^\top\mathbf{X}) \quad \text{s.t.} \quad \mathrm{rank}(\mathbf{M})\leq k\,. \tag{17}$$

We now consider the change of variable $\mathbf{M}' = \mathbf{M}\left(\tfrac{1}{n}\overline{\mathbf{X}}^\top\overline{\mathbf{X}}+\mathbf{\Sigma}\right)^{\frac{1}{2}}$. The above problem then becomes

$$\min_{\mathbf{M}'\in\mathbb{R}^{d\times d}} \quad \mathrm{Tr}(\mathbf{M}'\mathbf{M}'^\top) - 2\mathrm{Tr}\left(\mathbf{M}'\left(\tfrac{1}{n}\overline{\mathbf{X}}^\top\overline{\mathbf{X}}+\mathbf{\Sigma}\right)^{-\frac{1}{2}}\overline{\mathbf{X}}^\top\mathbf{X}\right) \quad \text{s.t.} \quad \mathrm{rank}(\mathbf{M}')\leq k\,. \tag{18}$$

This problem is equivalent to

$$\min_{\mathbf{M}'\in\mathbb{R}^{d\times d}} \quad \left\|\mathbf{M}' - \tfrac{1}{n}\mathbf{X}^\top\overline{\mathbf{X}}(\tfrac{1}{n}\overline{\mathbf{X}}^\top\overline{\mathbf{X}}+\mathbf{\Sigma})^{-\frac{1}{2}}\right\|_F^2 \quad \text{s.t.} \quad \mathrm{rank}(\mathbf{M}')\leq k\,. \tag{19}$$

Therefore the optimal $\mathbf{M}'$ is the Euclidean projection of $\tfrac{1}{n}\mathbf{X}^\top\overline{\mathbf{X}}\left(\tfrac{1}{n}\overline{\mathbf{X}}^\top\overline{\mathbf{X}}+\mathbf{\Sigma}\right)^{-\frac{1}{2}}$ onto the set of matrices of rank at most $k$. This projection has a well-known closed-form solution [23]. Consider the SVD of $\tfrac{1}{n}\mathbf{X}^\top\overline{\mathbf{X}}\left(\tfrac{1}{n}\overline{\mathbf{X}}^\top\overline{\mathbf{X}}+\mathbf{\Sigma}\right)^{-\frac{1}{2}}$:

$$\tfrac{1}{n}\mathbf{X}^\top\overline{\mathbf{X}}\left(\tfrac{1}{n}\overline{\mathbf{X}}^\top\overline{\mathbf{X}}+\mathbf{\Sigma}\right)^{-\frac{1}{2}} = \mathbf{R}\mathbf{\Phi}\mathbf{P}^\top \tag{20}$$

where $\mathbf{R}\in\mathbb{R}^{d\times d}$ is an orthogonal matrix, $\mathbf{\Phi}\in\mathbb{R}^{d\times d}$ is a diagonal matrix with singular values $\phi_1\geq\cdots\geq\phi_d\geq 0$ on the diagonal, and $\mathbf{P}\in\mathbb{R}^{d\times d}$ is an orthogonal matrix. The optimal $\mathbf{M}'$ is then given by:

$$\mathbf{M}'^\star = \mathbf{R}_k\mathbf{\Phi}_k\mathbf{P}_k^\top\,, \tag{21}$$

where $\mathbf{R}_k\in\mathbb{R}^{d\times k}$ and $\mathbf{P}_k\in\mathbb{R}^{d\times k}$ contain the first $k$ columns of $\mathbf{R}$ and $\mathbf{P}$, respectively, and $\mathbf{\Phi}_k\in\mathbb{R}^{k\times k}$ is a diagonal matrix with the first $k$ largest singular values $\phi_1,\ldots,\phi_k$ on the diagonal.

Then the optimal $\mathbf{M}$ is recovered as

$$\mathbf{M}^\star = \mathbf{M}'^\star\left(\tfrac{1}{n}\overline{\mathbf{X}}^\top\overline{\mathbf{X}}+\mathbf{\Sigma}\right)^{-\frac{1}{2}} = \mathbf{R}_k\mathbf{\Phi}_k\mathbf{P}_k^\top\left(\tfrac{1}{n}\overline{\mathbf{X}}^\top\overline{\mathbf{X}}+\mathbf{\Sigma}\right)^{-\frac{1}{2}}\,. \tag{22}$$

It follows that the optimal decoder $\mathbf{D}$ and encoder $\mathbf{E}$ are given by

$$\mathbf{D}^\star = \mathbf{R}_k\mathbf{\Phi}_k\mathbf{T}^{-1} \tag{23}$$

$$\mathbf{E}^\star = \mathbf{T}\mathbf{P}_k^\top\left(\tfrac{1}{n}\overline{\mathbf{X}}^\top\overline{\mathbf{X}}+\mathbf{\Sigma}\right)^{-\frac{1}{2}}\,, \tag{24}$$

where $\mathbf{T}$ is any invertible matrix in $\mathbb{R}^{k\times k}$. $\qquad\square$

## A.2   Proof of Theorem 3.2

**Theorem 3.2.** *Let* $\mathbf{S} := \tfrac{1}{n}\sum_i \mathbb{E}_{\tau\sim\mathcal{T}}\left[\tau(\mathbf{x}_i)\tau(\mathbf{x}_i)^\top\right]$, $\mathbf{G} := \tfrac{1}{n}\sum_i \mathbb{E}_{\tau\sim\mathcal{T}}\left[\tau(\mathbf{x}_i)\right]\mathbb{E}_{\tau\sim\mathcal{T}}\left[\tau(\mathbf{x}_i)\right]^\top$. *Assume that* $\mathbf{S}$ *is positive definite. Consider the eigendecomposition:*

$$\mathbf{S}^{-\frac{1}{2}}\mathbf{G}\mathbf{S}^{-\frac{1}{2}} = \mathbf{Q}\mathbf{\Omega}\mathbf{Q}^\top \tag{4}$$

*where* $\mathbf{\Omega} = \mathrm{diag}(\omega_1,\ldots,\omega_d)$ *with* $\omega_1\geq\cdots\geq\omega_d$. *Solutions of Equation* (SSL-JE) *for a linear model* $f_{\mathbf{W}}:\mathbf{x}\mapsto\mathbf{W}\mathbf{x}$ *take the form:*

$$\mathbf{W}^\star = \mathbf{U}\mathbf{Q}_k^\top\mathbf{S}^{-\frac{1}{2}}, \tag{5}$$

*where* $\mathbf{Q}_k = (\mathbf{q}_1,\ldots,\mathbf{q}_k)$ *and* $\mathbf{U}$ *is any orthogonal matrix of size* $k\times k$.

*Proof.* First, let us consider the constraint which ensures that the learned representations have orthonormal features. Using that $f_{\mathbf{W}} : \mathbf{x} \mapsto \mathbf{W}\mathbf{x}$ and $\mathbf{S} := \frac{1}{n}\mathbb{E}_{\tau \sim \mathcal{T}}\left[\tau(\mathbf{x}_i)\tau(\mathbf{x}_i)^\top\right]$, we obtain:

$$\frac{1}{n}\sum_{i \in [\![n]\!]} \mathbf{W}\,\mathbb{E}_{\tau \sim \mathcal{T}}\left[\tau(\mathbf{x}_i)\tau(\mathbf{x}_i)^\top\right]\mathbf{W}^\top = \mathbf{W}\mathbf{S}\mathbf{W}^\top = \mathbf{I}_k\,. \tag{25}$$

Then, the invariance term measures the consistency between positive views and is given by:

$$\frac{1}{2n}\sum_{i \in [\![n]\!]} \mathbb{E}_{\tau_1,\tau_2 \sim \mathcal{T}}\left[\|\mathbf{W}(\tau_1(\mathbf{x}_i) - \tau_2(\mathbf{x}_i))\|^2\right] = \mathrm{Tr}\left(\mathbf{W}\boldsymbol{\Sigma}\mathbf{W}^\top\right) \tag{26}$$

where

$$\boldsymbol{\Sigma} := \frac{1}{2n}\sum_{i \in [\![n]\!]} \mathbb{E}_{\tau_1,\tau_2 \sim \mathcal{T}}\left[(\tau_1(\mathbf{x}_i) - \tau_2(\mathbf{x}_i))(\tau_1(\mathbf{x}_i) - \tau_2(\mathbf{x}_i))^\top\right] \tag{27}$$

$$= \frac{1}{n}\sum_{i \in [\![n]\!]} \mathbb{E}_{\tau \sim \mathcal{T}}\left[\tau(\mathbf{x}_i)\tau(\mathbf{x}_i)^\top\right] - \mathbb{E}_{\tau \sim \mathcal{T}}\left[\tau(\mathbf{x}_i)\right]\mathbb{E}_{\tau \sim \mathcal{T}}\left[\tau(\mathbf{x}_i)\right]^\top \tag{28}$$

$$= \mathbf{S} - \mathbf{G}\,. \tag{29}$$

Thus we have

$$\mathrm{Tr}\left(\mathbf{W}\boldsymbol{\Sigma}\mathbf{W}^\top\right) = \mathrm{Tr}\left(\mathbf{W}\mathbf{S}\mathbf{W}^\top\right) - \mathrm{Tr}\left(\mathbf{W}\mathbf{G}\mathbf{W}^\top\right) \tag{30}$$

$$= \mathrm{Tr}\left(\mathbf{I}_k\right) - \mathrm{Tr}\left(\mathbf{W}\mathbf{G}\mathbf{W}^\top\right) \tag{31}$$

$$= k - \mathrm{Tr}\left(\mathbf{W}\mathbf{G}\mathbf{W}^\top\right)\,. \tag{32}$$

Therefore, the SSL problem simplifies to:

$$\max_{\mathbf{W} \in \mathbb{R}^{k \times d}}\quad \mathrm{Tr}\left(\mathbf{W}\mathbf{G}\mathbf{W}^\top\right) \tag{33}$$

$$\text{s.t.}\quad \mathbf{W}\mathbf{S}\mathbf{W}^\top = \mathbf{I}_k\,. \tag{34}$$

To solve this constrained non-convex optimization problem, we rely on the KKT conditions that are necessary conditions for optimality.

**First-order condition.** We introduce the Lagrangian:

$$\mathcal{L} = \mathrm{Tr}\left(\mathbf{W}\mathbf{G}\mathbf{W}^\top\right) - \mathrm{Tr}\left(\boldsymbol{\Lambda}\left(\mathbf{W}\mathbf{S}\mathbf{W}^\top - \mathbf{I}_k\right)\right)\,, \tag{35}$$

where $\boldsymbol{\Lambda} = \mathrm{diag}(\lambda_1, \ldots, \lambda_k)$ is the diagonal matrix of Lagrange multipliers. Taking the gradient of the Lagrangian with respect to $\mathbf{W}$ and setting it to zero yields:

$$\mathbf{W}^\star \mathbf{G} = \boldsymbol{\Lambda}\mathbf{W}^\star \mathbf{S}\,. \tag{36}$$

$\mathbf{S}$ is positive definite thus invertible and we can write the above as the following eigenvalue problem:

$$\mathbf{W}^\star \mathbf{G}\mathbf{S}^{-1} = \boldsymbol{\Lambda}\mathbf{W}^\star\,. \tag{37}$$

Since $\mathbf{S}$ is positive definite, it admits a unique positive definite square root $\mathbf{S}^{1/2}$, and we can write:

$$\mathbf{G}\mathbf{S}^{-1} = \mathbf{S}^{1/2}(\mathbf{S}^{-1/2}\mathbf{G}\mathbf{S}^{-1/2})\mathbf{S}^{-1/2}\,. \tag{38}$$

The matrix $\mathbf{S}^{-1/2}\mathbf{G}\mathbf{S}^{-1/2}$ is symmetric thus admits an eigendecomposition:

$$\mathbf{S}^{-1/2}\mathbf{G}\mathbf{S}^{-1/2} = \mathbf{Q}\boldsymbol{\Omega}\mathbf{Q}^\top, \tag{39}$$

where $\mathbf{Q}$ is orthogonal and $\boldsymbol{\Omega} = \mathrm{diag}(\omega_1, ..., \omega_d)$ with $\omega_1 \geq ... \geq \omega_d$. Substituting this into the expression for $\mathbf{G}\mathbf{S}^{-1}$, we obtain:

$$\mathbf{G}\mathbf{S}^{-1} = \mathbf{S}^{1/2}\mathbf{Q}\boldsymbol{\Omega}\mathbf{Q}^\top\mathbf{S}^{-1/2}. \tag{40}$$

Let $\mathbf{P} = \mathbf{S}^{1/2}\mathbf{Q}$. Then $\mathbf{G}\mathbf{S}^{-1}$ admits the decomposition:

$$\mathbf{G}\mathbf{S}^{-1} = \mathbf{P}\boldsymbol{\Omega}\mathbf{P}^{-1}. \tag{41}$$

In order to maximize the objective, the optimal choice for $\mathbf{W}^\star$ is to pick the eigenvectors associated with the $k$ largest eigenvalues of $\mathbf{G}\mathbf{S}^{-1}$. Therefore, the rows of $\mathbf{W}^\star$ lie in the span of $\{\mathbf{p}_1, \ldots, \mathbf{p}_k\}$. Precisely, there exists a matrix $\mathbf{U} \in \mathbb{R}^{k \times k}$ such that

$$\mathbf{W}^\star = \mathbf{U}\mathbf{P}_k^\top = \mathbf{U}\mathbf{Q}_k^\top\mathbf{S}^{-\frac{1}{2}} \tag{42}$$

where $\mathbf{P}_k = (\mathbf{p}_1, \ldots, \mathbf{p}_k)$ and $\mathbf{Q}_k = (\mathbf{q}_1, \ldots, \mathbf{q}_k)$.

**Primal feasibility.** Finally, solutions must satisfy the primal constraint: $\mathbf{W}^\star \mathbf{S} \mathbf{W}^{\star\top} = \mathbf{I}_k$. Using the first-order condition and the fact that $\mathbf{Q}_k^\top \mathbf{Q}_k = \mathbf{I}_k$, we obtain:

$$\mathbf{W}^\star \mathbf{S} \mathbf{W}^{\star\top} = \mathbf{U} \mathbf{Q}_k^\top \mathbf{S}^{-\frac{1}{2}} \mathbf{S} \mathbf{S}^{-\frac{1}{2}} \mathbf{Q}_k \mathbf{U}^\top = \mathbf{U} \mathbf{Q}_k^\top \mathbf{Q}_k \mathbf{U}^\top = \mathbf{U} \mathbf{U}^\top . \tag{43}$$

Therefore, to satisfy the condition $\mathbf{W}^\star \mathbf{S} \mathbf{W}^{\star\top} = \mathbf{I}_k$, it follows that $\mathbf{U}$ is an orthogonal matrix.

$\square$

## A.3 Proof of Proposition 4.2

**Proposition 4.2.** *[Supervised Learning] Let $\mathbf{V}^\star$ (resp. $\widetilde{\mathbf{V}}^\star$) be the linear model solving Equation* (SL) *with augmentation $\mathcal{T}(\alpha)$ for $\mathbf{X}$ (resp. the corrupted $\widetilde{\mathbf{X}}$). The limit:*

$$\widetilde{\mathbf{V}}^\star \xrightarrow[a.s.]{} \mathbf{V}^\star \tag{8}$$

*holds almost surely in either of the following regimes:*

- *as $\alpha \to +\infty$ (perfect augmentation-noise alignment) for any fixed sample size $n \in \mathbb{N}$.*

- *as $n \to +\infty$ (infinite samples) for any fixed alignment $\alpha \geq 0$.*

*Proof.* Let $\mathbf{Q} = (\mathbf{Q}_1 | \mathbf{Q}_2)$ where $\mathbf{Q}_1 \in \mathbb{R}^{d \times k}$ contains the first $k$ columns of $\mathbf{Q}$ and $\mathbf{Q}_2 \in \mathbb{R}^{d \times (d-k)}$ contains the remaining $d - k$ columns spanning the null directions of $\mathbf{X}$. All columns of $\mathbf{X}$ lie in the column space of $\mathbf{Q}_1$ and have no component along $\mathbf{Q}_2$ *i.e.* $\kappa_i > 0$ for $i \in [\![k]\!]$ and $\kappa_i = 0$ for $i > k$. Formally, $\mathbf{X} = \mathbf{X}_1 \mathbf{Q}_1^\top$ where $\mathbf{X}_1 = \mathbf{X} \mathbf{Q}_1 \in \mathbb{R}^{n \times k}$ has full column rank $k$ and $\mathbf{X} \mathbf{Q}_2 = \mathbf{0}$.

Recall that *noise components* are orthogonal to the column span of $\mathbf{Q}_1$. Indeed, $\mathbf{\Gamma} = \mathbf{Q} \mathbf{\Lambda}_{\mathbf{\Gamma}} \mathbf{Q}^\top$ with $\mathbf{\Lambda}_{\mathbf{\Gamma}} = \mathrm{diag}(\lambda_1^{\mathbf{\Gamma}}, \ldots, \lambda_d^{\mathbf{\Gamma}})$ such that $\lambda_i^{\mathbf{\Gamma}} = 0$ for any $i \in [\![k]\!]$. Finally, $\lambda_i^{\mathbf{\Gamma}} > 0$ for all $i \in [\![k+1 : d]\!]$.

**Uncorrupted data.** We consider the problem regularized by data augmentation given by Lemma B.1:

$$\min_{\mathbf{V} \in \mathbb{R}^{\ell \times d}} \frac{1}{n} \|\mathbf{Y} - \mathbf{X} \mathbf{V}^\top\|_F^2 + \|\mathbf{V}\|_{\mathbf{\Theta} + \alpha^2 \mathbf{\Gamma}}^2 . \tag{44}$$

Differentiating the objective with respect to $\mathbf{V}$ and setting the gradient to zero leads to the first-order optimality condition:

$$\frac{1}{n} (\mathbf{X} \mathbf{V}^{\star\top} - \mathbf{Y})^\top \mathbf{X} + \mathbf{V}^\star (\mathbf{\Theta} + \alpha^2 \mathbf{\Gamma}) = \mathbf{0}. \tag{45}$$

Given that the matrix $\mathbf{X}^\top \mathbf{X} + n(\mathbf{\Theta} + \alpha^2 \mathbf{\Gamma})$ is invertible, the closed form solution is given by:

$$\mathbf{V}^\star = \frac{1}{n} \mathbf{Y}^\top \mathbf{X} \left( \frac{1}{n} \mathbf{X}^\top \mathbf{X} + \mathbf{\Theta} + \alpha^2 \mathbf{\Gamma} \right)^{-1} . \tag{46}$$

Since $\mathbf{X} = \mathbf{X}_1 \mathbf{Q}_1^\top$ it gives:

$$\mathbf{V}^\star = \frac{1}{n} \mathbf{Y}^\top \mathbf{X}_1 \left( \frac{1}{n} \mathbf{X}_1^\top \mathbf{X}_1 + \mathbf{\Lambda}_{\mathbf{\Theta}, 1} \right)^{-1} \mathbf{Q}_1^\top \tag{47}$$

where $\mathbf{\Lambda}_{\mathbf{\Theta}, 1} = \mathrm{diag}(\lambda_1^{\mathbf{\Theta}}, \ldots, \lambda_k^{\mathbf{\Theta}})$ is the block of $\mathbf{\Lambda}_{\mathbf{\Theta}}$ corresponding to the subspace spanned by $\mathbf{Q}_1$.

**Corrupted data.** We now consider the corrupted data $\widetilde{\mathbf{X}} = \mathbf{X} + \mathbf{N}$ where $\mathbf{N} = (\mathbf{n}_1, \ldots, \mathbf{n}_n)^\top$. The $\{\mathbf{n}_i\}_{i \in [\![n]\!]}$ are independent variables such that for any $i \in [\![n]\!]$, $\mathbf{n}_i = \mathbf{\Gamma}^{\frac{1}{2}} \mathbf{z}_i$ where the $\{\mathbf{z}_i\}_{i \in [\![n]\!]}$ are independent $\mathcal{N}(\mathbf{0}, \mathbf{I}_d)$ vectors.

Using Lemma B.1 gives the following problem:

$$\min_{\widetilde{\mathbf{V}} \in \mathbb{R}^{\ell \times d}} \frac{1}{n} \|\mathbf{Y} - \widetilde{\mathbf{X}} \widetilde{\mathbf{V}}^\top\|_F^2 + \|\widetilde{\mathbf{V}}\|_{\mathbf{\Theta} + \alpha^2 \mathbf{\Gamma}}^2 . \tag{48}$$

Splitting $\widetilde{\mathbf{V}}$ into two parts: $\widetilde{\mathbf{V}}_1 = \widetilde{\mathbf{V}}\mathbf{Q}_1$ and $\widetilde{\mathbf{V}}_2 = \widetilde{\mathbf{V}}\mathbf{Q}_2$, one has that the optimal $\widetilde{\mathbf{V}}_1^\star$ is identical to the one in the uncorrupted case *i.e.*

$$\widetilde{\mathbf{V}}_1^\star = \tfrac{1}{n}\mathbf{Y}^\top\mathbf{X}_1 \left(\tfrac{1}{n}\mathbf{X}_1^\top\mathbf{X}_1 + \mathbf{\Lambda}_{\mathbf{\Theta},1}\right)^{-1} . \tag{49}$$

We define $\mathbf{N}_2 = \mathbf{N}\mathbf{Q}_2$. For $\widetilde{\mathbf{V}}_2$, canceling the gradient of the objective we get:

$$\tfrac{1}{n}\widetilde{\mathbf{V}}_2^\star\mathbf{N}_2^\top\mathbf{N}_2 - \tfrac{1}{n}\mathbf{Y}^\top\mathbf{N}_2 + \widetilde{\mathbf{V}}_2^\star(\mathbf{\Lambda}_{\mathbf{\Theta},2} + \alpha^2\mathbf{\Lambda}_{\mathbf{\Gamma},2}) = \mathbf{0} . \tag{50}$$

where $\mathbf{\Lambda}_{\mathbf{\Theta},2} = \text{diag}(\lambda_{k+1}^{\mathbf{\Theta}},\ldots,\lambda_d^{\mathbf{\Theta}})$ and $\mathbf{\Lambda}_{\mathbf{\Gamma},2} = \text{diag}(\lambda_{k+1}^{\mathbf{\Gamma}},\ldots,\lambda_d^{\mathbf{\Gamma}})$.

**a) Asymptotic $\alpha \to +\infty$.** In this regime, the above condition is equivalent to:

$$\widetilde{\mathbf{V}}_2^\star\mathbf{\Lambda}_{\mathbf{\Gamma},2} = \mathbf{0} \tag{51}$$

Since $\mathbf{\Lambda}_{\mathbf{\Gamma},2}$ is invertible, the first-order condition implies that $\widetilde{\mathbf{V}}_2^\star = \mathbf{0}$.

**b) Asymptotic $n \to +\infty$.** We obtain by the strong law of large numbers,

$$\tfrac{1}{n}\mathbf{Y}^\top\mathbf{N}_2 \xrightarrow[\text{a.s.}]{} \mathbf{0} \quad \text{and} \quad \tfrac{1}{n}\mathbf{N}_2^\top\mathbf{N}_2 \xrightarrow[\text{a.s.}]{} \mathbf{\Lambda}_{\mathbf{\Gamma},2} . \tag{52}$$

Hence at the limit we have,

$$\widetilde{\mathbf{V}}_2^\star(\mathbf{\Lambda}_{\mathbf{\Theta},2} + (1+\alpha^2)\mathbf{\Lambda}_{\mathbf{\Gamma},2}) = \mathbf{0} . \tag{53}$$

The matrix $\mathbf{\Lambda}_{\mathbf{\Theta},2} + (1+\alpha^2)\mathbf{\Lambda}_{\mathbf{\Gamma},2}$ is invertible for any $\alpha \geq 0$, thus it follows that $\widetilde{\mathbf{V}}_2^\star = \mathbf{0}$.

**Conclusion.** Therefore, both in the large-sample limit for a fixed $\alpha$ and in the large $\alpha$ limit for a fixed sample size, the optimal coefficients along the $\mathbf{Q}_2$-subspace vanish, *i.e.* $\widetilde{\mathbf{V}}_2^\star = \mathbf{0}$. Since $\widetilde{\mathbf{V}}_1^\star$ coincides with $\mathbf{V}_1^\star$, we recover the same solution as in the noiseless setting in both asymptotic regimes. $\qquad\square$

## A.4    Proof of Proposition 4.3

**Proposition 4.3.** *[Reconstruction] Let $\mathbf{E}^\star$ (resp. $\widetilde{\mathbf{E}}^\star$) be the linear (encoder) model solving Equation* (SSL-RC) *for $\mathbf{X}$ (resp. the corrupted $\widetilde{\mathbf{X}}$). The limit:*

$$\widetilde{\mathbf{E}}^\star \xrightarrow[\text{a.s.}]{} \mathbf{E}^\star \tag{9}$$

*holds[4] almost surely in either of the following regimes:*

- *as $\alpha \to +\infty$ (perfect augmentation-noise alignment) for any fixed sample size $n \in \mathbb{N}$.*

- *as $n \to +\infty$ (infinite samples), if and only if the alignment $\alpha \geq 0$ satisfies:*

$$\alpha^2 > \alpha_{\text{RC}}^2 := \max_{i\in[\![k+1:d]\!]} \frac{\lambda_i^{\mathbf{\Gamma}}}{\eta^2} - \frac{\lambda_i^{\mathbf{\Theta}}}{\lambda_i^{\mathbf{\Gamma}}} - 1 \quad \text{where} \quad \eta = \min_{i\in[\![k]\!]} \frac{\tfrac{1}{n}\kappa_i^2}{\sqrt{\tfrac{1}{n}\kappa_i^2 + \lambda_i^{\mathbf{\Theta}}}} . \tag{10}$$

*Proof.* Using Theorem 3.1, the closed-form solution for the encoder of the reconstruction SSL problem of Equation (SSL-RC) applied to $\mathbf{X}$ takes the form $\mathbf{E}^\star = \mathbf{T}\mathbf{P}_k^\top\left(\tfrac{1}{n}\mathbf{X}^\top\mathbf{X} + \mathbf{\Sigma}\right)^{-\frac{1}{2}}$ where $\mathbf{T}$ is any invertible matrix in $\mathbb{R}^{k\times k}$ and $\mathbf{P}_k$ is the matrix containing the $k$ columns of $\mathbf{P}$ associated with the $k$ largest singular values of the matrix $\tfrac{1}{n}\mathbf{X}^\top\mathbf{X}\left(\tfrac{1}{n}\mathbf{X}^\top\mathbf{X} + \mathbf{\Sigma}\right)^{-\frac{1}{2}}$. Given the construction of $\mathcal{T}(\alpha)$ (Section 4.1), if follows that $\mathbf{\Sigma} = \mathbf{\Theta} + \alpha^2\mathbf{\Gamma}$.

Let $\mathbf{Q} = (\mathbf{Q}_1|\mathbf{Q}_2)$ where $\mathbf{Q}_1 \in \mathbb{R}^{d\times k}$ contains the $k$ columns of $\mathbf{Q}$ corresponding to the *important components* and $\mathbf{Q}_2 \in \mathbb{R}^{d\times(d-k)}$ contains the remaining $d-k$ columns corresponding to the *noise components*. We denote $\mathbf{\Lambda}_{\mathbf{\Theta},1} = \text{diag}(\lambda_1^{\mathbf{\Theta}},\ldots,\lambda_k^{\mathbf{\Theta}})$, $\mathbf{\Lambda}_{\mathbf{\Theta},2} = \text{diag}(\lambda_{k+1}^{\mathbf{\Theta}},\ldots,\lambda_d^{\mathbf{\Theta}})$ and $\mathbf{\Lambda}_{\mathbf{\Gamma},2} = \text{diag}(\lambda_{k+1}^{\mathbf{\Gamma}},\ldots,\lambda_d^{\mathbf{\Gamma}})$.

---

[4]Up to an arbitrary invertible matrix (i.e., if $\mathbf{E}^\star$ is a solution, so is $\mathbf{T}\mathbf{E}^\star$ for any $k \times k$ invertible matrix $\mathbf{T}$).

**Uncorrupted data.** Decomposing $\frac{1}{n}\mathbf{X}^\top\mathbf{X} + \boldsymbol{\Sigma}$ and $\frac{1}{n}\mathbf{X}^\top\mathbf{X}$ in $(\mathbf{Q}_1|\mathbf{Q}_2)$ gives:

$$\frac{1}{n}\mathbf{X}^\top\mathbf{X} + \boldsymbol{\Sigma} = \mathbf{Q}_1\left(\frac{1}{n}\mathbf{K}_1^2 + \boldsymbol{\Lambda}_{\boldsymbol{\Theta},1}\right)\mathbf{Q}_1^\top + \mathbf{Q}_2\left(\boldsymbol{\Lambda}_{\boldsymbol{\Theta},2} + \alpha^2\boldsymbol{\Lambda}_{\boldsymbol{\Gamma},2}\right)\mathbf{Q}_2^\top \tag{54}$$

$$\frac{1}{n}\mathbf{X}^\top\mathbf{X} = \frac{1}{n}\mathbf{Q}_1\mathbf{K}_1^2\mathbf{Q}_1^\top . \tag{55}$$

Then using the same reasoning as in the proof of Proposition 4.4 (Appendix A.5) for the uncorrupted data case, we have that the eigenvalues of $\frac{1}{n}\mathbf{X}^\top\mathbf{X}\left(\frac{1}{n}\mathbf{X}^\top\mathbf{X} + \boldsymbol{\Sigma}\right)^{-\frac{1}{2}}$ on the *noise components* $\mathbf{Q}_2$ are all null and they are strictly positive on the *important components*. Therefore the largest eigenvalues are found on the *important components* $\mathbf{Q}_1$ and we obtain $\mathbf{P}_k = \mathbf{Q}_1$.

The solution of the reconstruction SSL problem is then given by

$$\mathbf{E}^\star = \mathbf{T}\left(\frac{1}{n}\mathbf{K}_1^2 + \boldsymbol{\Lambda}_{\boldsymbol{\Theta},1}\right)^{-\frac{1}{2}}\mathbf{Q}_1^\top \tag{56}$$

where $\mathbf{T}$ is any invertible matrix of size $k \times k$.

**Corrupted data.** Decomposing $\frac{1}{n}\widetilde{\mathbf{X}}^\top\widetilde{\mathbf{X}} + \boldsymbol{\Sigma}$ and $\frac{1}{n}\widetilde{\mathbf{X}}^\top\widetilde{\mathbf{X}}$ in $(\mathbf{Q}_1|\mathbf{Q}_2)$ gives:

$$\frac{1}{n}\widetilde{\mathbf{X}}^\top\widetilde{\mathbf{X}} + \boldsymbol{\Sigma} = \mathbf{Q}_1\left(\frac{1}{n}\mathbf{K}_1^2 + \boldsymbol{\Lambda}_{\boldsymbol{\Theta},1}\right)\mathbf{Q}_1^\top + \mathbf{Q}_2\left(\frac{1}{n}\mathbf{N}_2^\top\mathbf{N}_2 + \boldsymbol{\Lambda}_{\boldsymbol{\Theta},2} + \alpha^2\boldsymbol{\Lambda}_{\boldsymbol{\Gamma},2}\right)\mathbf{Q}_2^\top \tag{57}$$

$$\frac{1}{n}\widetilde{\mathbf{X}}^\top\widetilde{\mathbf{X}} = \frac{1}{n}\mathbf{Q}_1\mathbf{K}_1^2\mathbf{Q}_1^\top + \frac{1}{n}\mathbf{Q}_2\mathbf{N}_2^\top\mathbf{N}_2\mathbf{Q}_2^\top . \tag{58}$$

**a) Asymptotic $\alpha \to +\infty$.** In this asymptotic regime the term $\alpha^2\boldsymbol{\Lambda}_{\boldsymbol{\Gamma},2}$ dominates thus

$$\mathbf{Q}_2^\top\frac{1}{n}\widetilde{\mathbf{X}}^\top\widetilde{\mathbf{X}}\left(\frac{1}{n}\widetilde{\mathbf{X}}^\top\widetilde{\mathbf{X}} + \boldsymbol{\Sigma}\right)^{-\frac{1}{2}}\mathbf{Q}_2 \to \mathbf{0} . \tag{59}$$

Therefore the singular values of $\frac{1}{n}\widetilde{\mathbf{X}}^\top\widetilde{\mathbf{X}}\left(\frac{1}{n}\widetilde{\mathbf{X}}^\top\widetilde{\mathbf{X}} + \boldsymbol{\Sigma}\right)^{-\frac{1}{2}}$ on the *noise components* $\mathbf{Q}_2$ all converge to 0 almost surely. On the *important components* $\mathbf{Q}_1$, the smallest singular value is positive. Hence at the limit $\alpha \to +\infty$, we have $\mathbf{P}_k = \mathbf{Q}_1$.

**b) Asymptotic $n \to +\infty$.** Again using the strong law of large numbers we obtain

$$\frac{1}{n}\mathbf{N}_2^\top\mathbf{N}_2 \xrightarrow[\text{a.s.}]{} \boldsymbol{\Lambda}_{\boldsymbol{\Gamma},2} . \tag{60}$$

We denote by

$$\eta = \min_{i\in[\![k]\!]} \frac{\frac{1}{n}\kappa_i^2}{\sqrt{\frac{1}{n}\kappa_i^2 + \lambda_i^{\boldsymbol{\Theta}}}} . \tag{61}$$

In this asymptotic regime, ensuring that all eigenvalues of $\widetilde{\mathbf{X}}^\top\widetilde{\mathbf{X}}\left(\frac{1}{n}\widetilde{\mathbf{X}}^\top\widetilde{\mathbf{X}} + \boldsymbol{\Sigma}\right)^{-\frac{1}{2}}$ on *noise components* $\mathbf{Q}_2$ are smaller than eigenvalues on *important components* $\mathbf{Q}_1$ boils down to

$$\forall i \in [\![k+1:d]\!], \quad \eta > \frac{\lambda_i^{\boldsymbol{\Gamma}}}{\sqrt{\lambda_i^{\boldsymbol{\Theta}} + (1+\alpha^2)\lambda_i^{\boldsymbol{\Gamma}}}} . \tag{62}$$

Rearranging this inequality gives

$$\alpha^2 > \max_{i\in[\![k+1:d]\!]} \frac{\lambda_i^{\boldsymbol{\Gamma}}}{\eta^2} - \frac{\lambda_i^{\boldsymbol{\Theta}}}{\lambda_i^{\boldsymbol{\Gamma}}} - 1 . \tag{63}$$

Thus, if this condition is satisfied, we obtain $\mathbf{P}_k = \mathbf{Q}_1$.

**Conclusion.** In the large $\alpha$ limit for a fixed sample size, and in the large-sample limit under the condition of Equation (10), the SSL reconstruction problem is solved by

$$\widetilde{\mathbf{E}}^\star = \mathbf{T}\left(\frac{1}{n}\mathbf{K}_1^2 + \boldsymbol{\Lambda}_{\boldsymbol{\Theta},1}\right)^{-\frac{1}{2}}\mathbf{Q}_1^\top = \mathbf{E}^\star , \tag{64}$$

where $\mathbf{T}$ is any invertible matrix of size $k \times k$.

$\square$

## A.5 Proof of Proposition 4.4

**Proposition 4.4.** *[Joint-Embedding] Let $\mathbf{W}^\star$ (resp. $\widetilde{\mathbf{W}}^\star$) be the linear model solving Equation* (SSL-JE) *for $\mathbf{X}$ (resp. the corrupted $\widetilde{\mathbf{X}}$). The limit:*

$$\widetilde{\mathbf{W}}^\star \xrightarrow[a.s.]{} \mathbf{W}^\star \tag{11}$$

*holds[5] almost surely in either of the following regimes:*

- *as $\alpha \to +\infty$ (perfect augmentation-noise alignment) for any fixed sample size $n \in \mathbb{N}$.*

- *as $n \to +\infty$ (infinite samples), if and only if the alignment $\alpha \geq 0$ satisfies:*

$$\alpha^2 > \alpha_{\mathrm{JE}}^2 \coloneqq \max_{i \in [\![k+1:d]\!]} \frac{1-\delta}{\delta} - \frac{\lambda_i^{\boldsymbol{\Theta}}}{\lambda_i^{\boldsymbol{\Gamma}}} \quad where \quad \delta = \min_{i \in [\![k]\!]} \frac{\frac{1}{n}\kappa_i^2}{\frac{1}{n}\kappa_i^2 + \lambda_i^{\boldsymbol{\Theta}}} \ . \tag{12}$$

*Proof.* Let $\mathbf{Q} = (\mathbf{Q}_1 | \mathbf{Q}_2)$ where $\mathbf{Q}_1 \in \mathbb{R}^{d \times k}$ contains the $k$ columns of $\mathbf{Q}$ corresponding to the *important components* and $\mathbf{Q}_2 \in \mathbb{R}^{d \times (d-k)}$ contains the remaining $d-k$ columns corresponding to the *noise components*.

Using Theorem 3.2, the closed-form solution to the joint-embedding SSL problem of Equation (SSL-JE) applied to $\mathbf{X}$ takes the form $\mathbf{W}^\star = \mathbf{U}\mathbf{Q}_k^\top \mathbf{S}^{-\frac{1}{2}}$, where $\mathbf{U}$ is any orthogonal matrix of size $k \times k$. Recall that $\mathbf{Q}_k$ contains the $k$ columns of $\mathbf{Q}$ associated with the $k$ eigenvectors with largest eigenvalues of the matrix $\mathbf{S}^{-\frac{1}{2}}\mathbf{G}\mathbf{S}^{-\frac{1}{2}}$. Given the construction of $\mathcal{T}(\alpha)$ (Section 4.1), if follows that $\mathbf{S} = \frac{1}{n}\mathbf{X}^\top\mathbf{X} + \boldsymbol{\Theta} + \alpha^2\boldsymbol{\Gamma}$ and $\mathbf{G} = \frac{1}{n}\mathbf{X}^\top\mathbf{X}$.

We denote $\boldsymbol{\Lambda}_{\boldsymbol{\Theta},1} = \mathrm{diag}(\lambda_1^{\boldsymbol{\Theta}}, \ldots, \lambda_k^{\boldsymbol{\Theta}})$, $\boldsymbol{\Lambda}_{\boldsymbol{\Theta},2} = \mathrm{diag}(\lambda_{k+1}^{\boldsymbol{\Theta}}, \ldots, \lambda_d^{\boldsymbol{\Theta}})$ and $\boldsymbol{\Lambda}_{\boldsymbol{\Gamma},2} = \mathrm{diag}(\lambda_{k+1}^{\boldsymbol{\Gamma}}, \ldots, \lambda_d^{\boldsymbol{\Gamma}})$.

**Uncorrupted data.** Decomposing $\mathbf{S}$ and $\mathbf{G}$ in $(\mathbf{Q}_1 | \mathbf{Q}_2)$ gives:

$$\mathbf{S} = \mathbf{Q}_1 \left(\tfrac{1}{n}\mathbf{K}_1^2 + \boldsymbol{\Lambda}_{\boldsymbol{\Theta},1}\right) \mathbf{Q}_1^\top + \mathbf{Q}_2 \left(\boldsymbol{\Lambda}_{\boldsymbol{\Theta},2} + \alpha^2\boldsymbol{\Lambda}_{\boldsymbol{\Gamma},2}\right) \mathbf{Q}_2^\top \tag{65}$$

$$\mathbf{G} = \tfrac{1}{n}\mathbf{Q}_1\mathbf{K}_1^2\mathbf{Q}_1^\top \ . \tag{66}$$

It holds

$$\mathbf{Q}_1^\top \mathbf{S}^{-\frac{1}{2}}\mathbf{G}\mathbf{S}^{-\frac{1}{2}}\mathbf{Q}_2 = \mathbf{0}_{d-k} \ . \tag{67}$$

Hence on the *noise components* $\mathbf{Q}_2$, the eigenvalues of $\mathbf{S}^{-\frac{1}{2}}\mathbf{G}\mathbf{S}^{-\frac{1}{2}}$ are all null.

On the *important components* $\mathbf{Q}_1$, the smallest eigenvalue of $\mathbf{S}^{-\frac{1}{2}}\mathbf{G}\mathbf{S}^{-\frac{1}{2}}$ satisfies

$$\max_{i \in [\![k]\!]} \tfrac{1}{n}\kappa_i^2 \left(\tfrac{1}{n}\kappa_i^2 + \lambda_i^{\boldsymbol{\Theta}}\right)^{-1} > 0 \ , \tag{68}$$

since $\kappa_i > 0$ for any $i \in [\![k]\!]$. Therefore in the clean data setting, the largest eigenvalues of $\mathbf{S}^{-\frac{1}{2}}\mathbf{G}\mathbf{S}^{-\frac{1}{2}}$ are found on the *important component* thus it follows that $\mathbf{Q}_k = \mathbf{Q}_1$.

The solution of the joint-embedding SSL problem is then given by

$$\mathbf{W}^\star = \mathbf{U} \left(\tfrac{1}{n}\mathbf{K}_1^2 + \boldsymbol{\Lambda}_{\boldsymbol{\Theta},1}\right)^{-\frac{1}{2}} \mathbf{Q}_1^\top \tag{69}$$

where $\mathbf{U}$ is any orthogonal matrix of size $k \times k$.

---

[5]Up to an arbitrary orthogonal rotation (i.e., if $\mathbf{W}^\star$ is a solution, so is $\mathbf{U}\mathbf{W}^\star$ for any $k \times k$ orthogonal matrix $\mathbf{U}$).

**Corrupted data.** Let us denote $\mathbf{N}_2 = \mathbf{N}\mathbf{Q}_2$ with $\mathbf{N}$ being a $n \times d$ matrix whose rows are $\mathbf{n}_i \in \mathbb{R}^d$, where each $\mathbf{n}_i$ is drawn independently from $\mathcal{N}(\mathbf{0}, \boldsymbol{\Gamma})$.

We define

$$\widetilde{\mathbf{S}} := \frac{1}{n} \sum_{i \in [\![n]\!]} \mathbb{E}_{\tau \sim \mathcal{T}} \left[ \tau(\tilde{\mathbf{x}}_i) \tau(\tilde{\mathbf{x}}_i)^\top \right] , \tag{70}$$

$$\widetilde{\mathbf{G}} := \frac{1}{n} \sum_{i \in [\![n]\!]} \mathbb{E}_{\tau \sim \mathcal{T}} \left[ \tau(\tilde{\mathbf{x}}_i) \right] \mathbb{E}_{\tau \sim \mathcal{T}} \left[ \tau(\tilde{\mathbf{x}}_i) \right]^\top \tag{71}$$

which is the equivalent of matrix $\mathbf{S}$ when replacing the clean dataset $\mathbf{X}$ by the noisy dataset $\widetilde{\mathbf{X}}$. Decomposing $\widetilde{\mathbf{S}}$ and $\widetilde{\mathbf{G}}$ in $(\mathbf{Q}_1|\mathbf{Q}_2)$ gives:

$$\widetilde{\mathbf{S}} = \mathbf{Q}_1 \left( \tfrac{1}{n} \mathbf{K}_1^2 + \boldsymbol{\Lambda}_{\boldsymbol{\Theta},1} \right) \mathbf{Q}_1^\top + \mathbf{Q}_2 \left( \tfrac{1}{n} \mathbf{N}_2^\top \mathbf{N}_2 + \boldsymbol{\Lambda}_{\boldsymbol{\Theta},2} + \alpha^2 \boldsymbol{\Lambda}_{\boldsymbol{\Gamma},2} \right) \mathbf{Q}_2^\top \tag{72}$$

$$\widetilde{\mathbf{G}} = \tfrac{1}{n} \mathbf{Q}_1 \mathbf{K}_1^2 \mathbf{Q}_1^\top + \tfrac{1}{n} \mathbf{Q}_2 \mathbf{N}_2^\top \mathbf{N}_2 \mathbf{Q}_2^\top . \tag{73}$$

**a) Asymptotic $\alpha \to +\infty$.** In this asymptotic regime the term $\alpha^2 \boldsymbol{\Lambda}_{\boldsymbol{\Gamma},2}$ dominates thus

$$\mathbf{Q}_2^\top \widetilde{\mathbf{S}}^{-\frac{1}{2}} \widetilde{\mathbf{G}} \widetilde{\mathbf{S}}^{-\frac{1}{2}} \mathbf{Q}_2 \to \mathbf{0} . \tag{74}$$

Therefore the eigenvalues of $\widetilde{\mathbf{S}}^{-\frac{1}{2}} \widetilde{\mathbf{G}} \widetilde{\mathbf{S}}^{-\frac{1}{2}}$ on the *noise components* $\mathbf{Q}_2$ all converge to 0 almost surely. On the *important components* $\mathbf{Q}_1$, using the same argument as in the above uncorrupted data case, the smallest eigenvalue is strictly greater than 0. Therefore at the limit $\alpha \to +\infty$, the largest eigenvalues are found on the *important components* and we obtain $\mathbf{Q}_k = \mathbf{Q}_1$.

**b) Asymptotic $n \to +\infty$.** By the strong law of large numbers,

$$\tfrac{1}{n} \mathbf{N}_2^\top \mathbf{N}_2 \xrightarrow[\text{a.s.}]{} \boldsymbol{\Lambda}_{\boldsymbol{\Gamma},2} . \tag{75}$$

Let us denote

$$\delta = \min_{i \in [\![k]\!]} \frac{\tfrac{1}{n} \kappa_i^2}{\tfrac{1}{n} \kappa_i^2 + \lambda_i^{\boldsymbol{\Theta}}} . \tag{76}$$

In this asymptotic regime, ensuring that all eigenvalues of $\widetilde{\mathbf{S}}^{-\frac{1}{2}} \widetilde{\mathbf{G}} \widetilde{\mathbf{S}}^{-\frac{1}{2}}$ on *noise components* $\mathbf{Q}_2$ are smaller than eigenvalues on *important components* $\mathbf{Q}_1$ gives the condition:

$$\forall i \in [\![k+1:d]\!], \quad \delta > \frac{\lambda_i^{\boldsymbol{\Gamma}}}{\lambda_i^{\boldsymbol{\Theta}} + (1 + \alpha^2)\lambda_i^{\boldsymbol{\Gamma}}} . \tag{77}$$

Rearranging this inequality, we obtain

$$\alpha^2 > \max_{i \in [\![k+1:d]\!]} \frac{1 - \delta}{\delta} - \frac{\lambda_i^{\boldsymbol{\Theta}}}{\lambda_i^{\boldsymbol{\Gamma}}} . \tag{78}$$

Thus, if this condition is satisfied, the eigenvalues corresponding to the noise components are strictly smaller than those on the important components. Consequently, the $k$ largest eigenvalues of $\widetilde{\mathbf{S}}^{-\frac{1}{2}} \widetilde{\mathbf{G}} \widetilde{\mathbf{S}}^{-\frac{1}{2}}$ come solely from the data subspace *i.e.* $\mathbf{Q}_k = \mathbf{Q}_1$.

**Conclusion.** Both in the large $\alpha$ limit for a fixed sample size, and in the large-sample limit under the condition given by Equation (12), we obtain that the optimal solution of the SSL problem takes the form:

$$\widetilde{\mathbf{W}}^\star = \mathbf{U} \left( \tfrac{1}{n} \mathbf{K}_1^2 + \boldsymbol{\Lambda}_{\boldsymbol{\Theta},1} \right)^{-\frac{1}{2}} \mathbf{Q}_1^\top = \mathbf{W}^\star , \tag{79}$$

where $\mathbf{U}$ is any orthogonal matrix of size $k \times k$. Therefore, in these two regimes the solution has the same form as in the uncorrupted data setting.

$\square$

## A.6 Proof of Corollary 4.5

**Corollary 4.5.** *Let $\alpha_{\mathrm{JE}}$, $\delta$, $\alpha_{\mathrm{RC}}$, and $\eta$ be defined as in Proposition 4.4 and Proposition 4.3.*

- *If $\max_{i \in [\![k+1:d]\!]} \lambda_i^{\mathbf{\Gamma}} < \dfrac{\eta^2}{\delta}$ (low noise), then $\alpha_{\mathrm{JE}} > \alpha_{\mathrm{RC}}$ (reconstruction is preferable).*

- *If $\min_{i \in [\![k+1:d]\!]} \lambda_i^{\mathbf{\Gamma}} > \dfrac{\eta^2}{\delta}$ (high noise), then $\alpha_{\mathrm{JE}} < \alpha_{\mathrm{RC}}$ (joint-embedding is preferable).*

*Proof.* Recall the definition of $\alpha_{\mathrm{JE}}$ and $\alpha_{\mathrm{RC}}$:

$$\alpha_{\mathrm{JE}}^2 := \max_{i \in [\![k+1:d]\!]} \frac{1-\delta}{\delta} - \frac{\lambda_i^{\mathbf{\Theta}}}{\lambda_i^{\mathbf{\Gamma}}} \quad \text{where} \quad \delta = \min_{i \in [\![k]\!]} \frac{\frac{1}{n}\kappa_i^2}{\frac{1}{n}\kappa_i^2 + \lambda_i^{\mathbf{\Theta}}} , \tag{80}$$

$$\alpha_{\mathrm{RC}}^2 := \max_{i \in [\![k+1:d]\!]} \frac{\lambda_i^{\mathbf{\Gamma}}}{\eta^2} - \frac{\lambda_i^{\mathbf{\Theta}}}{\lambda_i^{\mathbf{\Gamma}}} - 1 \quad \text{where} \quad \eta = \min_{i \in [\![k]\!]} \frac{\frac{1}{n}\kappa_i^2}{\sqrt{\frac{1}{n}\kappa_i^2 + \lambda_i^{\mathbf{\Theta}}}} . \tag{81}$$

A sufficient condition for $\alpha_{\mathrm{JE}} > \alpha_{\mathrm{RC}}$ is the following, for any $i \in [\![k+1:d]\!]$:

$$\frac{1-\delta}{\delta} > \frac{\lambda_i^{\mathbf{\Gamma}}}{\eta^2} - 1 . \tag{82}$$

Rearranging this inequality gives:

$$\forall i \in [\![k+1:d]\!], \quad \lambda_i^{\mathbf{\Gamma}} < \frac{\eta^2}{\delta} . \tag{83}$$

Then, a sufficient condition for $\alpha_{\mathrm{JE}} < \alpha_{\mathrm{RC}}$ is the following inequality, for any $i \in [\![k+1:d]\!]$:

$$\frac{1-\delta}{\delta} < \frac{\lambda_i^{\mathbf{\Gamma}}}{\eta^2} - 1 . \tag{84}$$

It gives:

$$\forall i \in [\![k+1:d]\!], \quad \lambda_i^{\mathbf{\Gamma}} > \frac{\eta^2}{\delta} . \tag{85}$$

$\square$

# B  Effect of Data Augmentation for Supervised Learning

We recall a well-known result that establishes the equivalence between the effect of data augmentation and ridge regularization. We provide the proof for completeness.

**Lemma B.1.** *[7, 46] For any $\mathbf{V} \in \mathbb{R}^{\ell \times d}$, it holds:*

$$\frac{1}{n} \sum_{i \in [\![n]\!]} \mathbb{E}_{\tau \sim \mathcal{T}} \left[ \|\mathbf{y}_i - \mathbf{V}\tau(\mathbf{x}_i)\|_2^2 \right] = \|\mathbf{V}\|_{\mathbf{\Sigma}}^2 + \frac{1}{n} \sum_{i \in [\![n]\!]} \|\mathbf{y}_i - \mathbf{V}\mathbb{E}_{\tau \sim \mathcal{T}} \left[ \tau(\mathbf{x}_i) \right] \|_2^2 , \tag{86}$$

*where*

$$\mathbf{\Sigma} := \frac{1}{n} \sum_{i \in [\![n]\!]} \mathbb{E}_{\tau \sim \mathcal{T}} \left[ \tau(\mathbf{x}_i)\tau(\mathbf{x}_i)^\top \right] - \mathbb{E}_{\tau \sim \mathcal{T}} \left[ \tau(\mathbf{x}_i) \right] \mathbb{E}_{\tau \sim \mathcal{T}} \left[ \tau(\mathbf{x}_i) \right]^\top . \tag{87}$$

*Proof.*

$$\frac{1}{n} \sum_{i \in [\![n]\!]} \mathbb{E}_{\tau \sim \mathcal{T}} \left[ \|\mathbf{y}_i - \mathbf{V}\tau(\mathbf{x}_i)\|_2^2 \right] \tag{88}$$

$$= \frac{1}{n} \sum_{i \in [\![n]\!]} \|\mathbf{y}_i\|_2^2 + \mathbb{E}_{\tau \sim \mathcal{T}} \left[ \|\mathbf{V}\tau(\mathbf{x}_i)\|_2^2 \right] - 2\mathbb{E}_{\tau \sim \mathcal{T}} \left[ \text{Tr} \left( \mathbf{y}_i^\top \mathbf{V}\tau(\mathbf{x}_i) \right) \right] \tag{89}$$

$$= \frac{1}{n} \sum_{i \in [\![n]\!]} \|\mathbf{y}_i\|_2^2 + \text{Tr} \left( \mathbf{V}\mathbb{E}_{\tau \sim \mathcal{T}} \left[ \tau(\mathbf{x}_i)\tau(\mathbf{x}_i)^\top \right] \mathbf{V}^\top \right) - 2\text{Tr} \left( \mathbf{y}_i^\top \mathbf{V}\mathbb{E}_{\tau \sim \mathcal{T}} \left[ \tau(\mathbf{x}_i) \right] \right) \tag{90}$$

$$= \text{Tr} \left( \mathbf{V}\mathbf{\Sigma}\mathbf{V}^\top \right) + \frac{1}{n} \sum_{i \in [\![n]\!]} \|\mathbf{y}_i\|_2^2 - 2\text{Tr} \left( \mathbf{y}_i^\top \mathbf{V}\mathbb{E}_{\tau \sim \mathcal{T}} \left[ \tau(\mathbf{x}_i) \right] \right) + \|\mathbf{V}\mathbb{E}_{\tau \sim \mathcal{T}} \left[ \tau(\mathbf{x}_i) \right] \|_2^2 \tag{91}$$

$$= \|\mathbf{V}\|_{\mathbf{\Sigma}}^2 + \frac{1}{n} \sum_{i \in [\![n]\!]} \|\mathbf{y}_i - \mathbf{V}\mathbb{E}_{\tau \sim \mathcal{T}} \left[ \tau(\mathbf{x}_i) \right] \|_2^2 . \tag{92}$$

$\square$

## C   Experiments with Linear Models

In this section, we experiment with linear models and synthetic noise in order to validate the theoretical results of Section 4.

To process the data, we apply a PCA of dimension 50. We then add 50 components of noise to create a corrupted version of the dataset.

The eigenvalues for $\mathbf{\Gamma}$ and $\mathbf{\Theta}$ (as defined in Section 4.1) are randomly sampled from uniform distributions. Specifically, eigenvalues for $\mathbf{\Gamma}$ are drawn from the range $[0, \lambda_{\max}^{\mathbf{\Gamma}}]$, and eigenvalues for $\mathbf{\Theta}$ are drawn from $[0, \lambda_{\max}^{\mathbf{\Theta}}]$.

Across all experiments, we set a constant value for $\lambda_{\max}^{\mathbf{\Theta}} = 10^4$. To investigate the impact of input noise levels on model performance, we vary the value of $\lambda_{\max}^{\mathbf{\Gamma}}$. In the experiments presented in Figure 3, we define a weak noise case with $\lambda_{\max}^{\mathbf{\Gamma}} = 10^3$ and a strong noise case with $\lambda_{\max}^{\mathbf{\Gamma}} = 10^6$.

Joint-embedding and reconstruction solutions are obtained using the closed forms provided in Theorems 3.1 and 3.2. For evaluation, we compute the supervised linear probing score:

$$\min_{\mathbf{V} \in \mathbb{R}^{\ell \times k}} \frac{1}{n} \sum_{i \in [\![n]\!]} \|\mathbf{y}_i - \mathbf{V}\mathbf{z}_i\|_2^2 , \tag{93}$$

where $\mathbf{z}_i$ is the output of the SSL model on the $i$-th sample; *i.e.* $\mathbf{z}_i = \mathbf{W}^\star \mathbf{x}_i$ for joint embedding where $\mathbf{W}^\star$ is the optimal joint-embedding model of Theorem 3.2 and $\mathbf{z}_i = \mathbf{E}^\star \mathbf{x}_i$ for reconstruction where $\mathbf{E}^\star$ is the optimal encoder of the reconstruction model of Theorem 3.1. We then compute the absolute difference between the score of the model trained on clean data (*i.e.* , composed only of *important features*), and the score of the model trained on corrupted data (with the added *irrelevant noisy features*). As the model trained only on the *important features* naturally selects these features as SSL representations, this absolute difference directly quantifies the model's ability to filter out the *irrelevant noisy features*.

Figures 4, 5, 6, and 7 illustrate the experimental results, all of which support the intuitions outlined in Section 5.1. Notably, supervised models consistently discard noisy irrelevant components with increasing sample size or alignment strength, confirming the findings of Section A.3. However, SSL models demonstrate varied success depending on the setting. The *reconstruction* SSL model fails to retrieve *important components* from data with high noise magnitude, even with larger sample sizes and important alignment strength. Yet, with low noise, it successfully identifies these components and remains robust to alignment strength. In contrast, the *joint-embedding* SSL model requires a certain minimum alignment strength to filter out noisy irrelevant components, even in low-noise settings, but it exhibits strong robustness to increasing noise magnitude (bottom figures).

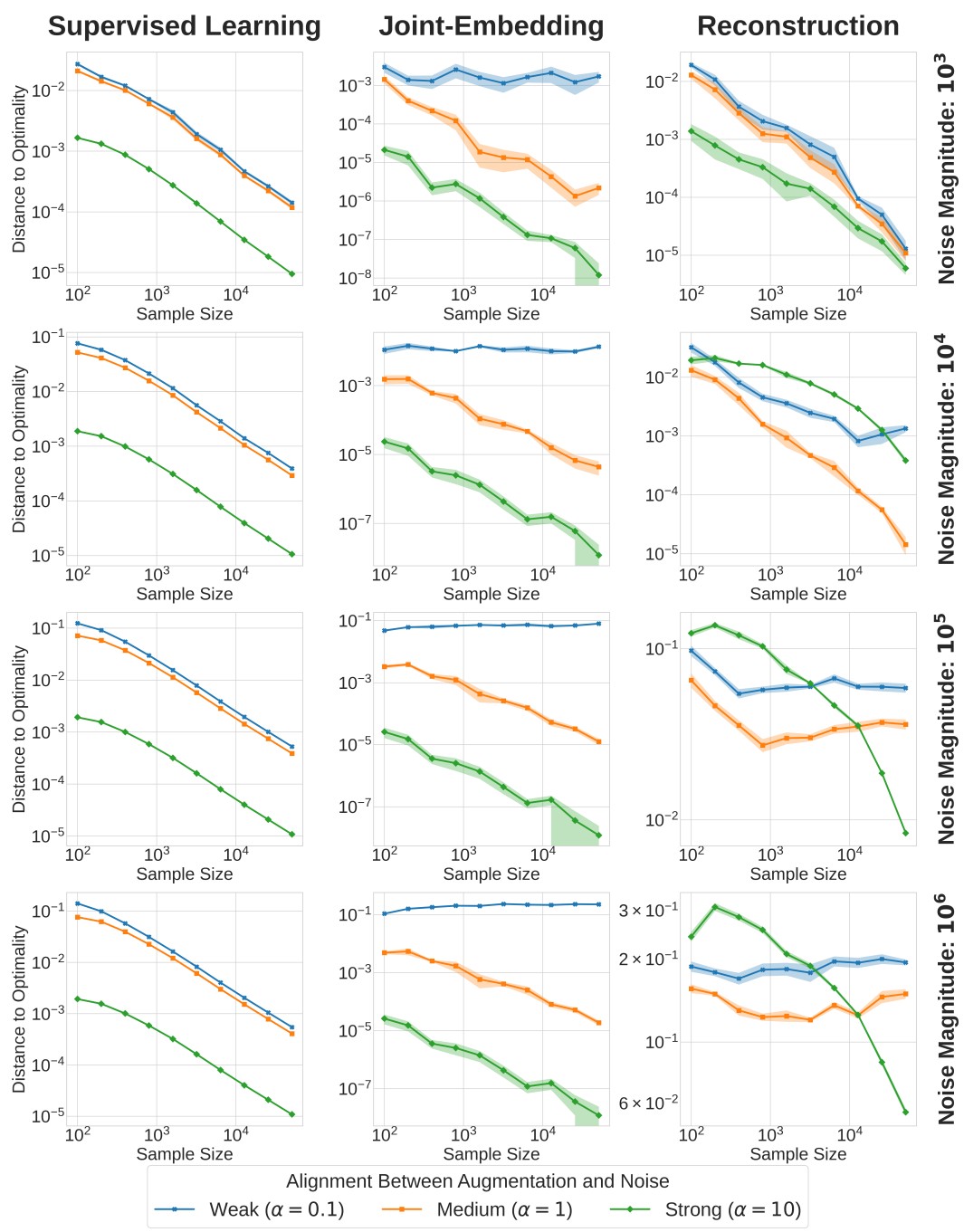

Figure 4: Performance of linear supervised and SSL models (Sections 3.1 and 3.2 and Theorems 3.1 and 3.2) with synthetic noise (Section 4.1 and Appendix C) on **MNIST**. Each subplot's y-axis is the absolute difference of supervised linear probing loss (on clean vs. corrupted data) and its x-axis is the sample size $n$.

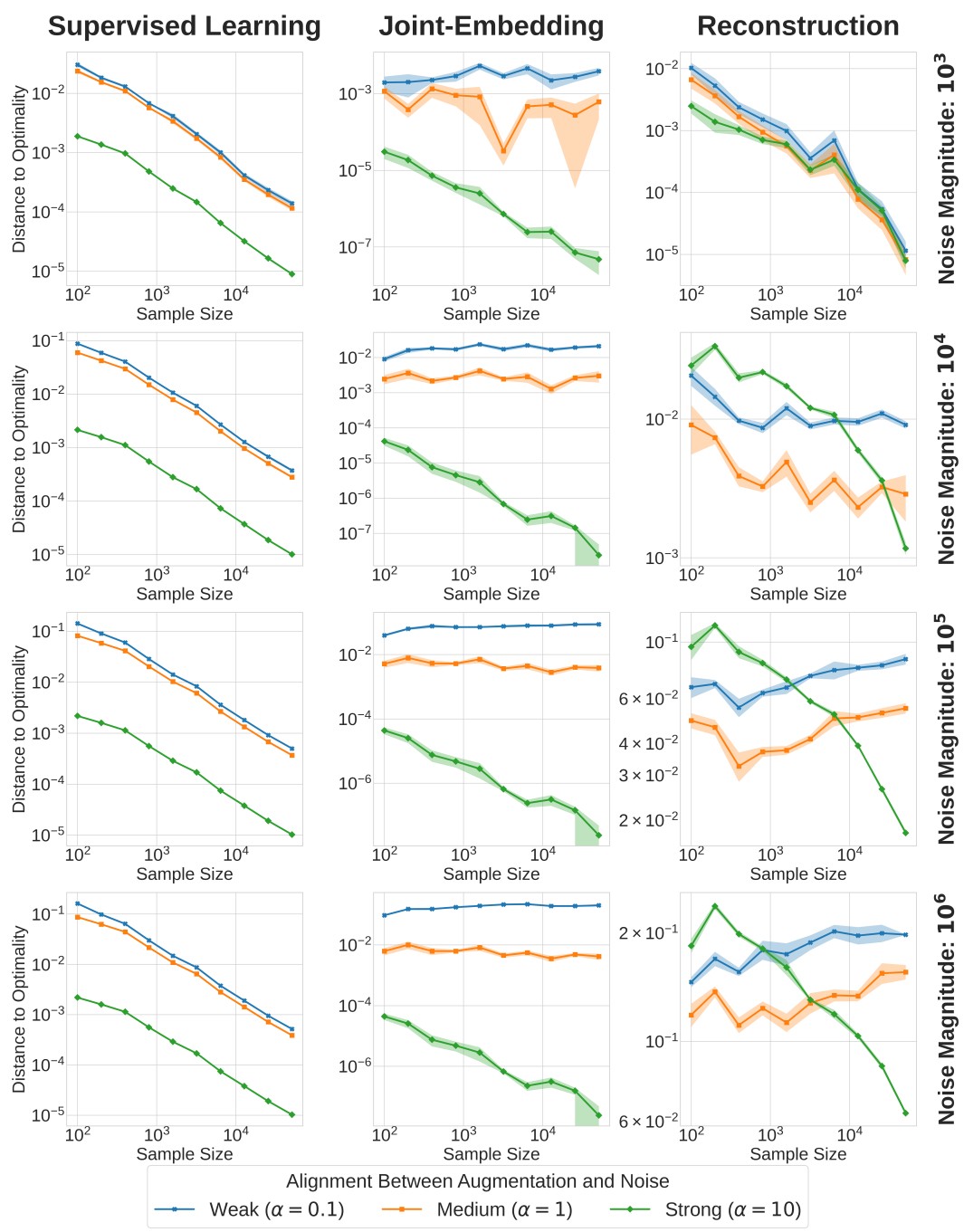

Figure 5: Performance of linear supervised and SSL models (Sections 3.1 and 3.2 and Theorems 3.1 and 3.2) with synthetic noise (Section 4.1 and Appendix C) on **Fashion-MNIST**. Each subplot's y-axis is the absolute difference of supervised linear probing loss (on clean vs. corrupted data) and its x-axis is the sample size $n$.

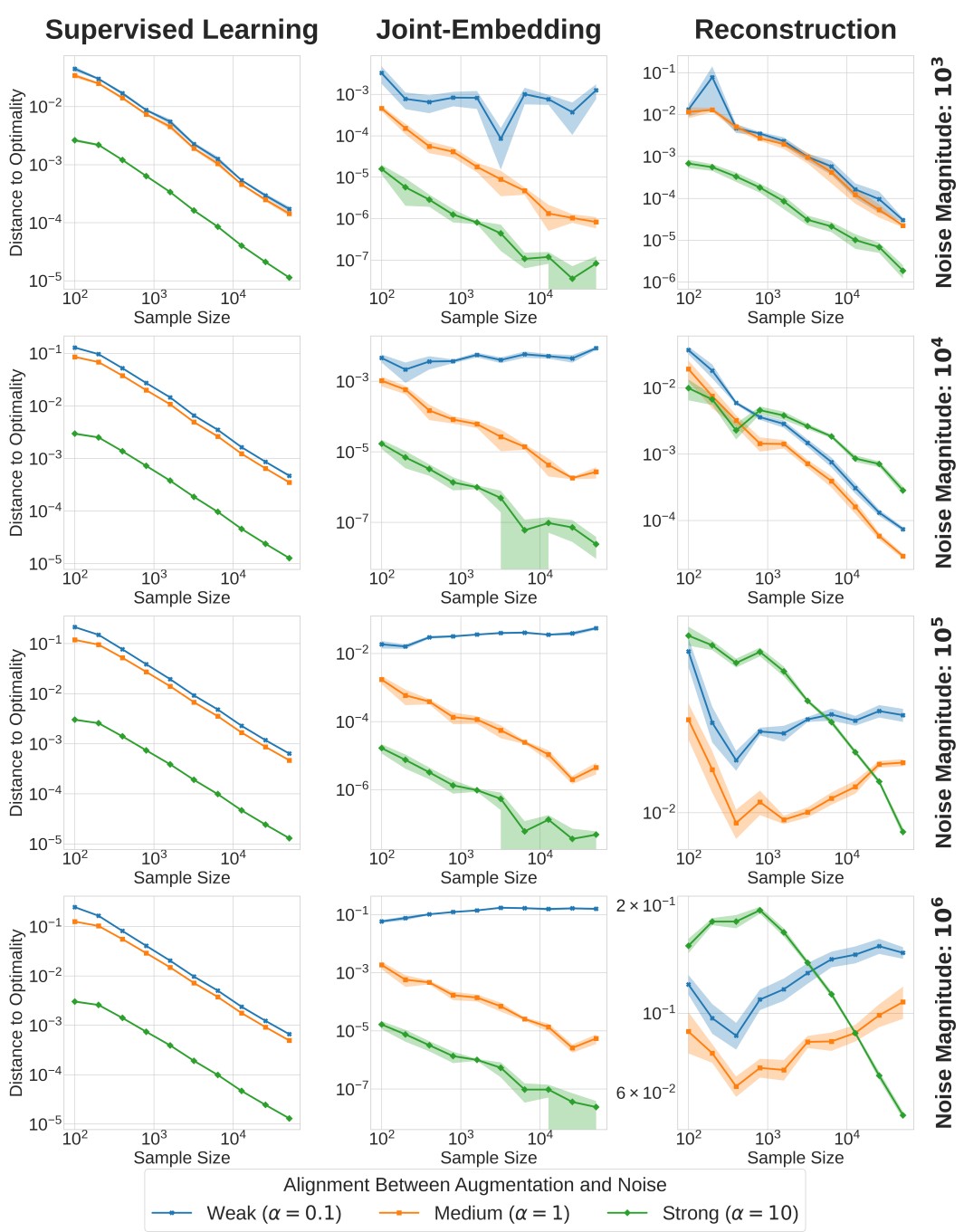

Figure 6: Performance of linear supervised and SSL models (Sections 3.1 and 3.2 and Theorems 3.1 and 3.2) with synthetic noise (Section 4.1 and Appendix C) on **Kuzushiji-MNIST** characters [16]. Each subplot's y-axis is the absolute difference of supervised linear probing loss (on clean vs. corrupted data) and its x-axis is the sample size $n$.

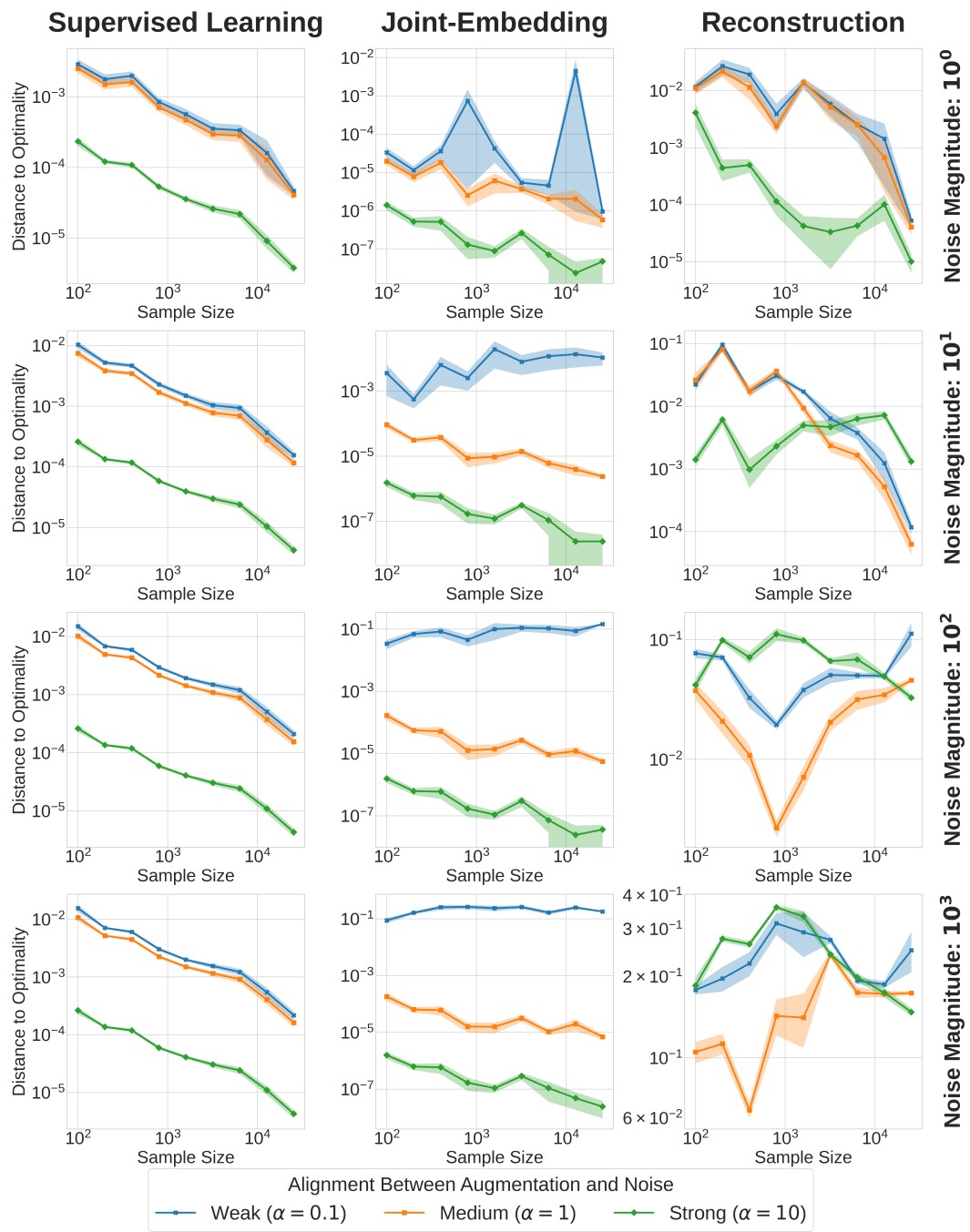

Figure 7: Performance of linear supervised and SSL models (Sections 3.1 and 3.2 and Theorems 3.1 and 3.2) with synthetic noise (Section 4.1 and Appendix C) on a **single-cell RNA seq** dataset from [49]. Each subplot's y-axis is the absolute difference of supervised linear probing loss (on clean vs. corrupted data) and its x-axis is the sample size $n$.

# D Experiments with Deep Networks on Image Classification

## D.1 Details about the Experiments

This section details the experimental setup and hyperparameters used in the experiments.

For the evaluation of SSL methods, we freeze the learned representations and then conduct linear probing, reporting the top-1 accuracy on the test set. For the data augmentation pipelines and SSL modules, we use default modules from the `lightly` library [55].

To introduce *irrelevant noisy features* in the data, we use the ImageNet-C corruptions [35]. This benchmark provides a comprehensive suite of corruptions designed to assess model robustness. ImageNet-C corruptions span categories such as noise, blur, weather, and digital distortions, each available at multiple levels of severity. For each experiment, we create a *fixed corrupted dataset* by assigning a deterministic corruption to each image, ensuring every time an image is loaded it undergoes the same corruption. This effectively simulates a corrupted dataset, consistent across epochs. For the CIFAR-10 dataset, these corruptions were adapted to the specific image dimensions.

### D.1.1 ImageNet Experiments

For the ImageNet [18] experiments, we use a batchsize of 256 and train the model for 500 epochs. We use a ResNet-50 backbone [34] for BYOL [26] with a learning rate of 0.45 and a weight decay of $10^{-6}$, with the LARS optimizer [69]. We use a cosine scheduler from 0.99 to 1 for the student momentum parameter. For MAE and DINO, we use a ViT-B/16 backbone [21], and the AdamW optimizer [48] with a learning rate of $1.5 \times 10^{-4}$ and a weight decay of 0.05.

For all methods, the hyperparameters, including architectural choice, were set to the values for ImageNet presented in their respective original papers. For augmentations, we use the default augmentations associated to BYOL, DINO and MAE from the lightly library [55] with default parameters.

### D.1.2 CIFAR10 Experiments

In Appendices D.2 and D.3, we perform experiments on the CIFAR-10 [42] dataset. For these experiments, we use a ResNet-50 backbone [34], a batch size of 256, and train the model for 1000 epochs. The LARS optimizer [69] is employed with a learning rate of 5 and a weight decay of $10^{-6}$. Supervised training is conducted using the same architecture and optimization parameters. All experiments are run with 5 different random seeds.

We conduct experiments using three SSL methods: SimCLR [14], BYOL [26], and VICReg [10]. For SimCLR, we set the temperature parameter to $\tau = 0.5$. For VICReg, we use the following default hyperparameters: a scaling coefficient of 25 for the invariance term of the loss, 25 for the variance term, and 1 for the covariance term. For BYOL, we use a cosine scheduler from 0.99 to 1 for the student momentum parameter. For augmentations, we use the default augmentations associated to SimCLR, BYOL and VICReg from the `lightly` library [55] adapted to the CIFAR-10 dataset as follows: the random resized crop is set to 32 and Gaussian blur is removed.

## D.2 Self-Supervised Learning is much more Sensitive to Corruptions than Supervised Learning

In this section, we compare the performances of SimCLR against a supervised model.

Scores for SimCLR and supervised learning with the same augmentations on corrupted datasets at various corruption strengths are shown in Figure 8. We observe that the performance of SimCLR tends to degrade rapidly as the level of noise in the data increases. A similar trend is observed for supervised learning, but the decline is significantly less steep. For instance, performance on corruptions such as fog and frost remains relatively stable across corruption strengths for supervised learning, whereas for SSL, performance can drop by a factor of two (e.g., from approximately 0.8 to 0.4 in top-1 accuracy).

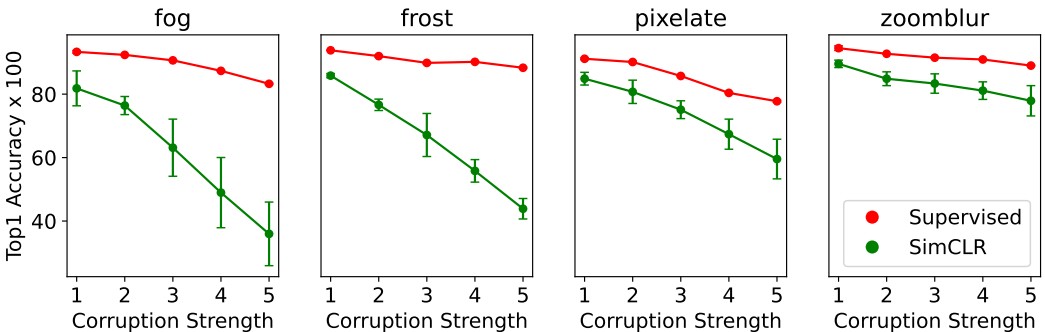

Figure 8: Top-1 accuracy for supervised learning and self-supervised learning methods on CIFAR-10 under various corruptions with severity levels ranging from 1 to 5. SSL is performed using SimCLR with default augmentations while supervised learning uses the same augmentations as SimCLR. We observe that supervised learning exhibits greater robustness to data corruption compared to SSL. This confirms the theoretical results of Section 4.

This phenomenon is confirmed by examining the learned representations of VICReg in Figure 2, without and with noisy corruptions, also on CIFAR10. While the supervised model maintains a clear separation of clusters even under noisy conditions, the VICReg representations degrade significantly and fail to distinguish clusters effectively when data is strongly corrupted.

These observations corroborate the results proved in Section 4. Indeed, the lack of labels prevents the model from compensating for any misalignment between augmentation and noise. Consequently, SSL performance deteriorates rapidly in the presence of corruptions if the data augmentations are not appropriately adapted.

### D.3   Aligning Augmentation with Known Noise

In this section, we validate the findings of Propositions 4.3 and 4.4 and demonstrate empirically the impact of aligning the data augmentation with the *irrelevant noisy features* in the data. To do so, we artificially create misalignment with usual augmentations by applying a known corruption to the dataset. We then investigate whether augmenting the data with the same type of noise can help the model learn better representations.

Specifically, for each view, we append a transform of the same corruption type but not necessarily the same strength, at the end of the data augmentation pipeline. We refer to the corruption applied to the dataset as the *Corruption Strength* and the additional corruption appended during augmentation as the *Augmentation Strength*. For each corruption tested, we evaluate the *Corruption Strength* over the set $\{1, 2, 3, 4, 5\}$ and the *Augmentation Strength* over the set $\{0, 1, 2\}$.

Experiments are conducted on CIFAR10 and results are displayed in Figure 9. For each heatmap, we are interested in verifying if the top two rows corresponding to *augmentation strength* = 1 and 2 yield better results than the bottom row without noise injection (*augmentation strength* = 0). Looking at SimCLR runs, we observe that *noise injection* generally boosts performance in most settings. This holds true for all 16 corruptions, except for spatter and brightness, where an augmentation strength of 0 performs best. Some corruptions clearly demonstrate that noise injection improves performance regardless of the corruption strength. For instance, this is evident for fog, Gaussian blur, and glass blur. For other corruptions, the utility of noise injection depends on the strength of the noise. For example, frost, saturate, and snow benefit from noise injection when the corruption in the input data is strong, whereas for impulse noise, JPEG compression, and defocus blur, noise injection in the augmentation pipeline is more effective when the corruption in the input data is not severe. In some cases, the improvements are substantial; for instance, for fog with a corruption strength of 4, adding noise injection with a strength of 2 increases the top-1 accuracy from 49.0 to 67.1. Across all configurations of noise and corruption strength, a total

of 80 combinations are tested with SimCLR, comprising 16 corruption types at 5 strength levels. Noise injection leads to an improvement in top-1 accuracy in 67.5% of these cases.

We observe that these trends are quite consistent across various SSL methods. For VICReg, noise injection improves top-1 accuracy in 85% of the configurations, while for BYOL, this improvement is observed in 60% of the tested configurations. For VICReg, noise injection results in significant improvements; for instance, in the case of fog with *corruption strength* = 5, the top-1 accuracy increases dramatically from 43.5 to 70.9. This performance improvement is evident in Figure 2, where the clusters corresponding to class labels are more clearly separated when noise injection is applied. For completeness, we provide in Figure 10 the top-1 linear probing accuracy scores evaluated on a clean test set (standard CIFAR-10 test set). We observe that for SimCLR, noise injection improves top-1 accuracy in 68.75% of the configurations, while for both BYOL and VICReg, it enhances performance in 85% of the configurations.

These outcomes confirm the results of Proposition 4.4. Aligning augmentations with noise directs the model's focus to important features, thereby enhancing SSL representations. Note that our theory does not cover cases where data and noise components are intertwined, as seen with *e.g.* spatter, where strong noise may discard important data features and render *noise injection* less effective. Finally, one key observation is that, even under substantial data corruption, a small injection of noise during augmentation can still yield benefits. Hence the augmentation noise strength does not need to be precisely tuned to match the severity of the corruption.

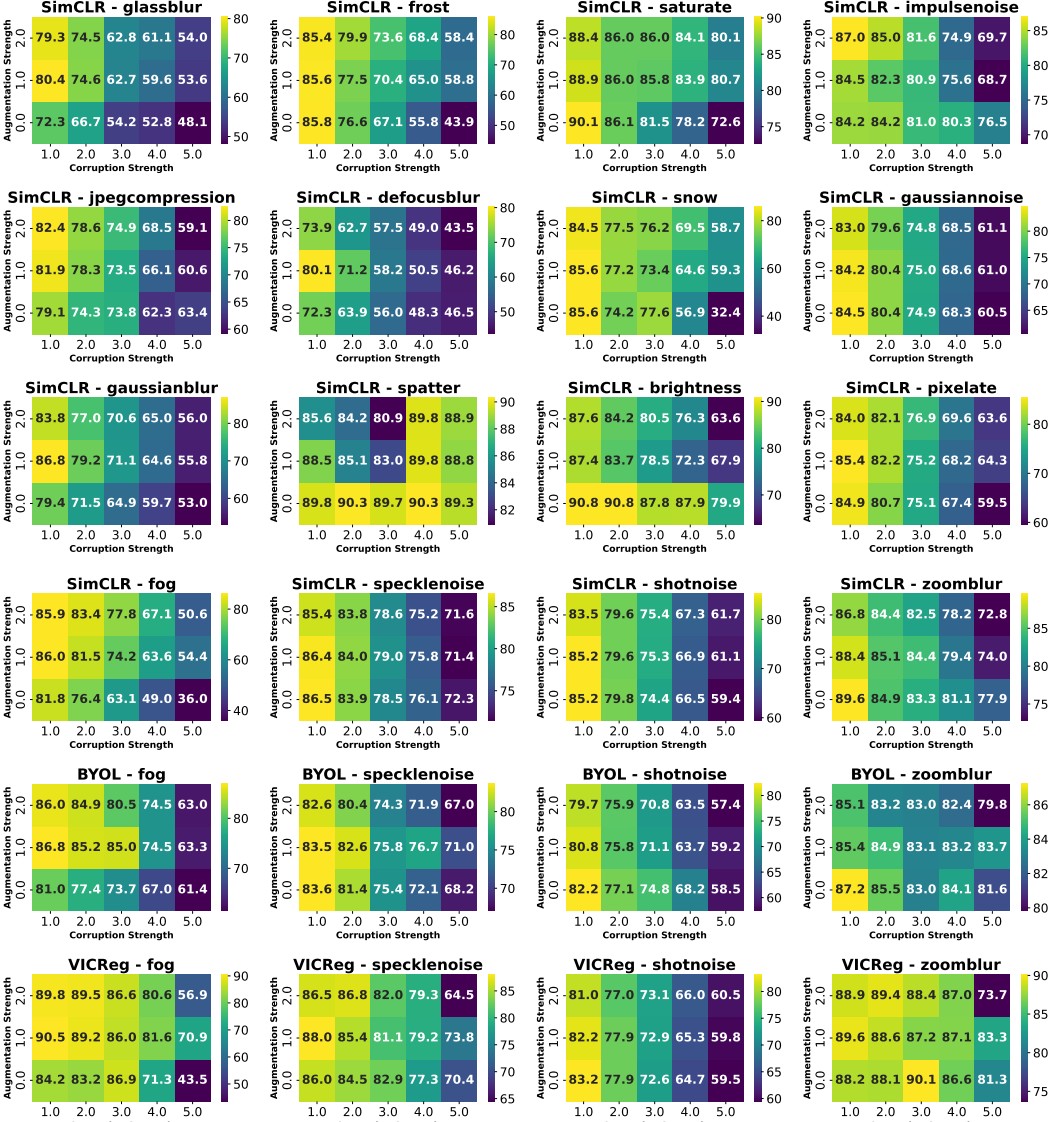

Figure 9: Top-1 linear probing accuracy of SSL methods evaluated on CIFAR10 test set with the same corruption type and severity as the training set. The experiments investigate varying levels of corruption severity in the input data (Corruption Strength) and different levels of noise injection severity in the augmentation pipeline (Augmentation Strength). The noise injection corruption type matches the data corruption type. Each reported value is an average over 5 random seeds. The first four rows present SimCLR results across 16 distinct corruptions: glassblur, frost, saturate, impulsenoise, jpegcompression, defocusblur, snow, gaussiannoise, gaussianblur, spatter, brightness, pixelate, fog, specklenoise, shotnoise and zoomblur. These transformations are sourced from Imagenet-C [35]. The bottom two rows display results for BYOL and VICReg on a subset of 4 corruptions: fog, specklenoise, shotnoise and zoomblur.

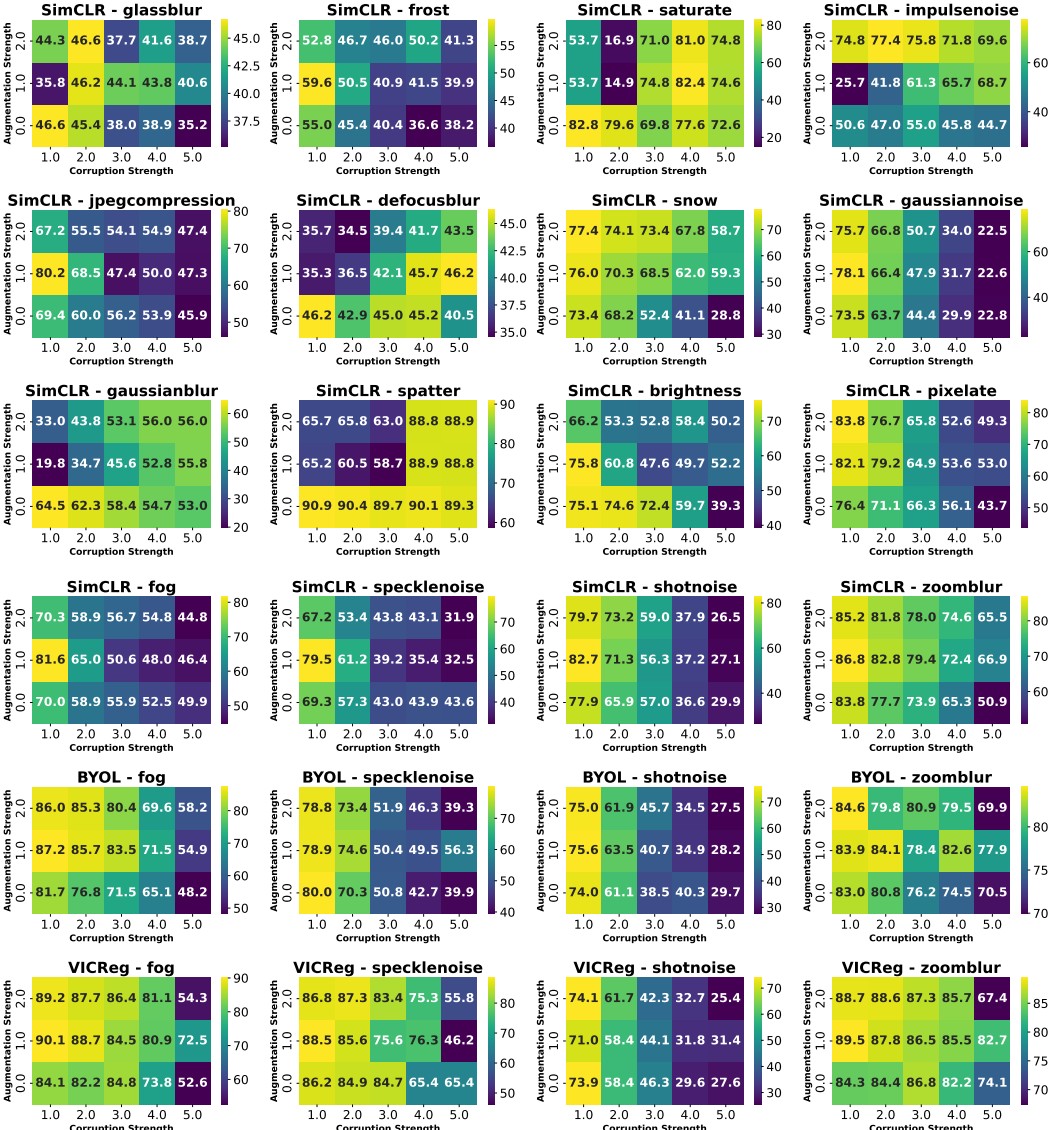

Figure 10: Top-1 linear probing accuracy for SSL methods evaluated on the clean (uncorrupted) CIFAR10 test set. The experiments investigate varying levels of corruption severity in the input data (Corruption Strength) and different levels of noise injection severity in the augmentation pipeline (Augmentation Strength). The noise injection corruption type matches the data corruption type. Each reported value is an average over 5 random seeds. The first four rows present SimCLR results across 16 distinct corruptions: glassblur, frost, saturate, impulsenoise, jpegcompression, defocusblur, snow, gaussiannoise, gaussianblur, spatter, brightness, pixelate, fog, specklenoise, shotnoise and zoomblur. These transformations are sourced from Imagenet-C [35]. The bottom two rows display results for BYOL and VICReg on a subset of 4 corruptions: fog, specklenoise, shotnoise and zoomblur.

