# OpenReview forum: "Joint‑Embedding vs Reconstruction: Provable Benefits of Latent Space Prediction for Self‑Supervised Learning"
_NeurIPS.cc/2025/Conference — NeurIPS 2025 spotlight_

### Official Review · Reviewer_ufsf · 2025-06-30

**Clarity:** 3
**Significance:** 2
**Originality:** 3
**Rating:** 4
**Confidence:** 4

**Summary:**

The authors compare two popular self-supervised learning (SSL) families—reconstruction (e.g., MAE) and joint-embedding (e.g., BYOL, SimCLR, BarlowTwins).
By analysing a simplified linear setting they prove that which family works better depends on how strongly “unwanted” input variations (noise, corruptions, background) show up in the data and how well the chosen augmentations imitate those variations. They then confirm the theory with small synthetic tasks and with ImageNet-C / CIFAR-10-C experiments.

**Questions:**

How could one estimate alignment or noise magnitude from raw, unlabelled data in practice?

Can the analysis be extended to non-Gaussian augmentations such as random crops, colour jitter or CutMix? In particular, how might you handle the patch-masking used in MAE?

Do you have intuition or experiments for settings where signal and nuisance features overlap in the same subspace?

**Ethical Concerns:**

["NO or VERY MINOR ethics concerns only"]

**Final Justification:**

The theoretical results are clearly stated, although there are some limitations. However, the authors acknowledged and addressed these limitations in the rebuttal phase.

**Limitations:**

No.
The paper never talks about what it can’t do or any downsides. It should briefly add:
Simplifying assumptions – It assumes linear models and simple Gaussian noise. Real data and augmentations are messier, so results may change.
Small-data limits – All proofs need huge sample sizes; no clue if things hold for typical datasets.
How to use it – Readers still don’t know how to measure “noise strength” or “alignment” in practice.

**Quality:**

3

**Strengths And Weaknesses:**

Strengths:
The research question is very practical and important: Both loss families are widely used, yet few papers ask which one to pick and when.
The theory is clean: A straightforward linearization yields transparent derivations and an easy-to-remember rule of thumb: if the unwanted variations are mild, reconstruction suffices; if they are strong, joint-embedding performs better.
The paper has decent empirical support:  Simple linear demos and modern vision models on corrupted ImageNet/CIFAR largely follow the predicted pattern.

Weaknesses:
The promised guidance for practitioners is thin. The paper never demonstrates how to estimate augmentation–noise alignment or noise strength on real, unlabelled data.
All proofs rely on additive Gaussian noise and scaled augmentations. Real augmentations (crops, colour jitter, CutMix, masking) are more complex, so it is unclear how well the conclusions generalise.
The theory assumes that signal and noise occupy orthogonal directions, an assumption that seldom holds for natural images or text, where the two are usually entangled.

---

> ### Author Rebuttal · Authors · 2025-07-25
>
> We would like to thank the reviewer for the insightful feedback and comments.
>
> > Can the analysis be extended to non-Gaussian augmentations such as random crops, color jitter or CutMix? In particular, how might you handle the patch-masking used in MAE?
>
> We appreciate the reviewer’s question regarding extending our Gaussian covariance-based perturbation analysis, giving us the opportunity to clarify an important point.
>
> Let us consider the first-order Taylor expansion around the mean augmentation. Defining $\bar{\tau}(\mathbf{x}) := \mathbb{E}_{\tau \sim \mathcal{T}}[\tau(\mathbf{x})]$, for a given function $f$, we approximate:
> $$
> f(\tau(\mathbf{x})) \approx f(\bar{\tau}(\mathbf{x})) + Df(\bar{\tau}(\mathbf{x}))[\delta\tau(\mathbf{x})],
> \quad\text{with}\quad
> \delta\tau(\mathbf{x}) = \tau(\mathbf{x}) - \bar{\tau}(\mathbf{x}).
> $$
> where $D$ is the Jacobian operator. From this linear approximation, the induced covariance of $f(\tau(\mathbf{x}))$ is given by:
> $$
> \mathrm{Cov}(f(\tau(\mathbf{x})))
> \approx Df(\bar{\tau}(\mathbf{x})) \mathbf{\Sigma} Df(\bar{\tau}(\mathbf{x}))^\top,
> \quad\text{where}\quad
> \mathbf{\Sigma} = \mathbb{E}[\delta\tau(\mathbf{x})\delta\tau(\mathbf{x})^\top].
> $$
> as defined in equation (Cov) in the paper. When $f$ is linear, as considered in this work for analytical tractability (consistent with the SSL literature focused on tractability [a][b]), this approximation becomes exact as no higher-order terms remain.
> In that case, we recover the same mean and covariance for $f(\tau(\mathbf{x}))$ as if we used the additive Gaussian noise model with covariance $\mathbf{\Sigma}$ to define $\tau$.
>
> Thus, modeling the augmentation distribution as Gaussian provides a natural second-order approximation, matching exactly the first two statistical moments. Notably, several practically relevant augmentations admit closed-form covariance expressions, as explicitly derived (see page 11 in [c]). These include Pepper noise injection, Random Cutout, and unbiased random masking. This shows that masking, in particular, can be captured through a tractable second-moment approximation, directly addressing the reviewer's question about its compatibility with our framework.
> Concretely, to incorporate such augmentations, one can simply substitute the corresponding closed-form covariance operator into our formalism (as Gaussian covariance), effectively mimicking the effect of the augmentation through its second-order moment.
>
> While finite-moment approximations of data augmentations and noise are established in supervised learning contexts [d], they remain under-explored in the SSL literature. Current analytical results in SSL [e] rarely exploit the structural properties of the data augmentation distribution explicitly, despite its central role. We believe leveraging finite-moment approximations is a promising direction for deriving meaningful theoretical insights for SSL methods.
>
> We will insist on these aspects in the revised version to clarify our approach.
>
> [a] Cabannes, Vivien, et al. "The ssl interplay: Augmentations, inductive bias, and generalization." ICML 2023
>
> [b] Etai Littwin, et al. "How jepa avoids noisy features: The implicit bias of deep linear self distillation networks." NeurIPS 2024
>
> [c] Lin, Chi-Heng, et al. "The good, the bad and the ugly sides of data augmentation: An implicit spectral regularization perspective." JMLR 25.91 (2024): 1-85.
>
> [d] Balestriero, R., I. Misra, and Y. LeCun. "A Data-Augmentation Is Worth A Thousand Samples: Exact Quantification From Analytical Augmented Sample Moments. NeurIPS 2022
>
> [e] Balestriero, Randall, and Yann LeCun. "Contrastive and non-contrastive self-supervised learning recover global and local spectral embedding methods." NeurIPS 2022
>
> > How could one estimate alignment or noise magnitude from raw, unlabelled data in practice?
>
> We thank the reviewer for this important question related to the point raised above.
>
> Our work provides precise analytical quantities in settings where second-order moment approximations hold and where noise statistics can be estimated from data (bounds of proposition 4.3 and 4.4). While the practical estimation of these quantities in real-world systems is indeed valuable, it falls beyond the scope of this study. Our focus is on establishing a controlled, theoretically grounded comparison between joint embedding and reconstruction-based SSL methods.
>
> Moreover, our experiments on real image datasets confirm the theoretical trend: reconstruction-based methods are significantly more sensitive to noise perturbations than joint embedding approaches, even beyond controlled or synthetic settings. This phenomenon has been observed in several prior studies, and our work provides a theoretical explanation for it. We believe this connection is valuable for the SSL community.
>
> > Do you have intuition or experiments for settings where signal and nuisance features overlap in the same subspace?
>
> Yes, we do cover such cases in our experiments. In Section 5, we evaluate models on corrupted images from ImageNet-C, including transformations such as Pixelate, Gaussian noise, and Zoom Blur. These corruptions affect the same subspace as core signal features.
>
> Despite this overlap, we observe the same consistent trend: joint embedding methods remain more robust than reconstruction-based methods as noise severity increases. This suggests that the robustness of joint embedding extends even to more challenging settings where signal and nuisance features are entangled.
>
> > Limitations
>
> We thank the reviewer for pointing out that the limitations of our work should be more clearly highlighted. We will address this in the revised version of the paper by more explicitly discussing the key assumptions and scope.
>
> We appreciate the thoughtful discussion and remain at your disposal to elaborate on any part of our submission.

---

> > ### Comment · Reviewer_ufsf · 2025-08-04
> >
> > I thank the authors for addressing my questions and concerns about the non-Gaussian augmentations and other limitations. I raised my point as a result.

---

### Official Review · Reviewer_YR7n · 2025-07-01

**Clarity:** 3
**Significance:** 4
**Originality:** 4
**Rating:** 4
**Confidence:** 3

**Summary:**

This paper investigates the differences in the learning dynamics and performance of reconstruction-based and joint embedding-based self-supervised approaches in an attempt to enable and guide practitioners to make an educated choice of one over the other.

**Questions:**

1) Are linear models used in section 3 a good proxy for the complex non-linear models that are used in practice? How do we ensure that the findings transfer to these more complex non-linear models?
2) I understand how increasing alpha in Eq. 7 results in a more aligned augmentation with the noise added. However, how do the authors control for “unaligned” noise? It seems to me that the extremes of alpha either result in a dominant aligned augmentation with high alpha or just the theta term with alpha as 0. Shouldn’t we also control for the magnitude of theta?
3) Propositions 4.2-4: How do we conclude that the limit holds (almost surely) as either alpha or n goes to positive infinity? It is unclear to me from the main text without any pointers to appendix.
4) Figure 2, bottom right (Strong noise, reconstruction). Do the authors have insight why the model with strong alignment lags behind at a low sample size despite high alignment between the noise and the augmentation?
5) For Table 1, I would highly suggest color-coding or adding a symbol to help readers separate joint embedding (BYOL, DINO) methods from reconstruction (MAE) methods at a glance.

**Ethical Concerns:**

["NO or VERY MINOR ethics concerns only"]

**Final Justification:**

I thank the authors for addressing my points, I have no further questions. I will maintain my original score.

**Limitations:**

yes

**Paper Formatting Concerns:**

No formatting concerns.

**Quality:**

3

**Strengths And Weaknesses:**

**Strengths**:
1) Well motivated and overall writing is good.
2) A variety of experiments backup authors’ claims.
3) Theoretical results consistent with and further explain results and patterns in previous work.
4) The work would equip researchers and practitioners to better be able to decide on approaches

**Weaknesses**:
1) Unclear exactly what a good alignment of noise and augmentation means in practice. As far as I could tell, in real-world datasets when the high-frequency details are somewhat irrelevant, joint-embedding techniques are better. However, how do we judge this on other datasets? How can we judge if specific augmentations would be effective?
2) Please refer to questions below.

---

> ### Author Rebuttal · Authors · 2025-07-29
>
> We would like to thank the reviewer for the constructive and insightful feedback.
>
> > 1. Are linear models used in section 3 a good proxy for the complex non-linear models that are used in practice? How do we ensure that the findings transfer to these more complex non-linear models?
>
> Thanks a lot for this important question. We appreciate the opportunity to clarify why we chose a linear‑parameter framework for our analysis.
>
> Modeling our system as linear in its parameters yields closed‑form solutions that can be derived and analyzed exactly. This approach is standard in theoretical studies of self‑supervised learning (SSL): for instance, prior work on the interplay of augmentations and inductive bias [a], on implicit bias in self‑distillation networks [b], and on SSL dynamics without contrastive pairs [c] all adopt linear‑parameter models to perform precise derivations.
>
> Importantly, “linear” in our work refers only to dependence on the parameters, not to the representation power of the model. We still allow arbitrary nonlinear feature maps, akin to kernel regression. Moreover, it is common to linearize around initialization via a first‑order Taylor expansion to study deep neural networks. In the infinite‑width limit, the Neural Tangent Kernel (NTK) framework [d] shows that gradient descent training behaves exactly like kernel regression with a fixed kernel, implying that the dynamics are linear in the parameters. Thus, our linear‑parameter analysis can be understood as capturing the same training regime as very wide deep networks.
>
> In supervised learning, recent work has derived closed‑form regularization effects of data augmentation [e]. Our contribution is the analogous analysis for SSL, where augmentations play an even more central role.
>
> Finally, as emphasized in [f], many fundamental aspects of (implicit or explicit) regularization can be understood in the linear‑model formalism. We believe that the fact that we can explain the key distinction between reconstruction‑based and joint‑embedding objectives with such a simple standard formalism highlights the generality and clarity of our findings.
>
> [a] Cabannes, Vivien, et al. "The ssl interplay: Augmentations, inductive bias, and generalization." ICML 2023
>
> [b] Etai Littwin, et al. "How jepa avoids noisy features: The implicit bias of linear self distillation networks." NeurIPS 2024
>
> [c] Tian, Yuandong, Xinlei Chen, and Surya Ganguli. "Understanding self-supervised learning dynamics without contrastive pairs." ICML 2021
>
> [d] Jacot, Arthur, Franck Gabriel, and Clément Hongler. "Neural tangent kernel: Convergence and generalization in neural networks." NeurIPS 2018.
>
> [e] Balestriero, Randall, Ishan Misra, and Yann LeCun. "A data-augmentation is worth a thousand samples: Analytical moments and sampling-free training." NeurIPS 2022.
>
> [f] Bach, Francis. Learning theory from first principles. MIT press, 2024.
>
>
> > 2. I understand how increasing alpha in Eq. 7 results in a more aligned augmentation with the noise added. However, how do the authors control for “unaligned” noise? It seems to me that the extremes of alpha either result in a dominant aligned augmentation with high alpha or just the theta term with alpha as 0. Shouldn’t we also control for the magnitude of theta?
>
> Thanks for the question. The covariance $\mathbf{\Theta}$ can be any positive semidefinite matrix, potentially arbitrarily far from the true noise covariance $\mathbf{\Gamma}$. Our theoretical results hold in full generality for any choice of $\mathbf{\Theta}$. Concretely, if we choose $\mathbf{\Theta}$ to assign large eigenvalues to irrelevant noise components and small eigenvalues to informative data components, then we reproduce the case of a large $\alpha$ where we match the data augmentation to the noise. However this is not the scenario we are interested in. This is why in the experiments we consider no correlation between the default augmentation structure ($\mathbf{\Theta}$) and the actual noise ($\mathbf{\Gamma}$).
>
> In our simulated experiments, we independently and uniformly sample the eigenvalue spectra of both $\mathbf{\Theta}$ and $\mathbf{\Gamma}$. This ensures there is no correlation between augmentation and noise covariances. Setting $\alpha = 0$ recovers the case where there is no correlation between actual noisy features and augmentation (because the augmentation follows the structure given by $\mathbf{\Theta}$), while larger $\alpha$ interpolates towards aligning the augmentation with the noise (discarding the effect of $\mathbf{\Theta}$ as $\alpha$ increases).
>
> This protocol also mirrors our real‑data image experiments, where we inject corruption noise (from Imagenet-C) not present in the standard augmentation pipeline. Here, $\mathbf{\Theta}$ represents the default augmentation pipeline, unaware of the injected noise, thus creating a natural mismatch.
>
>
> > 3. Propositions 4.2-4: How do we conclude that the limit holds (almost surely) as either alpha or n goes to positive infinity? It is unclear to me from the main text without any pointers to appendix.
>
> We apologize for omitting explicit pointers to the proofs in the main text. In the revised version, we will add direct links before Propositions 4.2–4 to guide readers to Appendix A.3–A.5.
>
> The proofs in Appendix A.3–A.5 proceed by first invoking the closed‑form solutions from Theorems 3.1 and 3.2. We then decompose each solution into two parts: the components that cover important data features and the other components that cover the irrelevant noise features. We then analyze each part in the limits of infinite sample size ($n\to\infty$) and perfect alignment ($\alpha\to\infty$). The infinite‑sample limit uses the law of large numbers to establish almost sure convergence of empirical covariances to their population counterparts, while the perfect‑alignment limit isolates the effect of the noise structure in the global augmentation covariance. We then compare the solutions obtained for clean and corrupted data settings.
>
> We will also refine our notation and expand intermediate steps in the appendix to further improve readability. Please let us know if you would like any additional details.
>
>
> > 4. Figure 2, bottom right (Strong noise, reconstruction). Do the authors have insight why the model with strong alignment lags behind at a low sample size despite high alignment between the noise and the augmentation?
>
> This is indeed a consistent observation and an interesting point also raised by reviewer CuYw. We provide the same answer below for consistency.
>
> We believe this discrepancy stems from finite‑sample effects. By the multivariate Central Limit Theorem, the variance of the empirical noise‑covariance spectrum decays like $1/\sqrt{n}$. In low‑sample regimes, this high variance causes the spectrum to deviate from the true noise covariance $\mathbf{\Gamma}$, thus worsening probing error. Because the excess variance scales with $\alpha^2$, its impact is greatest at large $\alpha$. As $n$ increases, these fluctuations vanish, and strong alignment consistently outperforms weaker settings, as predicted by Proposition 4.3.
> Importantly, we do not observe this behavior with joint embeddings: because both views undergo the same augmentation pipeline, they maintain identical variance levels for each $\alpha$, even in small‑sample scenarios.
>
> We will incorporate this discussion in the revised version, thanks a lot for pointing this out.
>
>
> > 5. For Table 1, I would highly suggest color-coding or adding a symbol to help readers separate joint embedding (BYOL, DINO) methods from reconstruction (MAE) methods at a glance.
>
> Thank you for this excellent suggestion. We will do this in the revised version to make the results clearer for readers.
>
> We’re grateful for the reviewers’ comments and are ready to provide additional clarification or expand on any point you wish.

---

> > ### Comment · Reviewer_YR7n · 2025-08-04
> >
> > I thank the authors for addressing my points, I have no further questions.

---

### Official Review · Reviewer_FFZ4 · 2025-07-02

**Clarity:** 3
**Significance:** 4
**Originality:** 3
**Rating:** 6
**Confidence:** 3

**Summary:**

In this work, the authors theoretically analyze a linear implementation of two well-known self-supervised learning paradigms: reconstruction and joint-embedding based. The theoretical analysis is made under the assumption of existing data augmentations, i.e., the existence of a distribution of transformations. This idea has been previously explored in supervised learning, where the results conclude that the quadratic risk minimization problem is equivalent to a Tikhonov regularized problem. The authors derive, using similar principles, closed-form solutions for reconstruction and joint-embedding problems, where the results clearly indicate the tendency of the optimal solutions to align with the principal axes of variation in the data. Finally, under asymptotic regimes of infinite data and augmentation alignment, they derive critical criteria regarding the alignment of the models with the signal components that are important for solving the task. Experiments on synthetic and real datasets (with well-known SSL  implementations) validate the theoretical conclusions.

**Questions:**

Please consult the weaknesses section above

**Ethical Concerns:**

["NO or VERY MINOR ethics concerns only"]

**Final Justification:**

After reading the rebuttal, I am maintaining my initial score. I believe that the authors have provided clarifications to both myself and the other reviewers regarding the practical relevance of the results, especially regarding the use of a quadratic loss instead of the typical categorical cross-entropy. Finally, I would like to thank the authors for the timely and detailed rebuttal.

**Limitations:**

yes

**Paper Formatting Concerns:**

no formatting issues

**Quality:**

4

**Strengths And Weaknesses:**

Strengths:
- The study presents the first attempt at characterizing the differences between joint-embedding and reconstruction/generative models in closed form under a distribution of augmentations. This is an interesting step ahead in a field that is becoming increasingly more relevant in practice.
- The closed form results for both paradigms are derived cleanly, and they align with intuition, for example a reconstruction based method will have an optimal encoder that captures data covariance and the covariance due to augmentations, project it on the top-k directions and then freely transform it (linearly) to maintain this k-rank projection. This corresponds with the intuition practitioners have regarding generative models as "closely fitting the details of the data distribution."
- The derivation of the two regimes of $\alpha$ and which type of model can perform better under different noise and augmentation regimes is an important result.
- The experiments, although few, confirm the theoretical results both on ideal (synthetic) and real-world cases. This is an optimal result, especially considering that the analysis is done on linear models and linear optimization principles.

Weaknesses and Questions:
- One of the main assumptions of the theoretical analysis is that the labels are considered to be continuous in order to frame a least squares task. While this obviously allows for an in-depth theoretical exploration, how would it affect the performance of SSL models downstream on categorical matching tasks?
- In Equation 7, where the augmentation distribution is formalized, is my understanding correct that the directions that are unimportant to the task are augmented more, i.e., there is more noise added, as alpha grows, in the directions with $\lambda_i^\Gamma = 0$. Is this correct?
- As a minor element and more of a suggestion, I believe the proofs would benefit more people looking to understand theoretical research in SSL if some of the identities, e.g., regarding matrix traces and norms, were expanded more in the first few derivations. This greatly helps the readability of the more technical material.

---

> ### Author Rebuttal · Authors · 2025-07-28
>
> We would like to thank the reviewer for the constructive review and very positive feedback.
>
> > 1. One of the main assumptions of the theoretical analysis is that the labels are considered to be continuous in order to frame a least squares task. While this obviously allows for an in-depth theoretical exploration, how would it affect the performance of SSL models downstream on categorical matching tasks?
>
> Indeed, probing is often done through discrete labels hence this question is very relevant for practical scenarios. We agree that treating labels as continuous values via MSE is primarily a theoretical convenience.
>
> In practice, MSE can be applied to classification by representing categorical labels via one-hot encoding, where each class corresponds to a unique basis vector. This approach enables the use of regression-based objectives even when the downstream task is inherently categorical.
> Importantly, recent empirical work [a] has demonstrated that training classifiers with squared loss on one-hot targets can match, and in some cases even exceed, the performance of models trained with cross-entropy loss. These findings suggest that squared loss is not only theoretically convenient but also practically viable for classification.
>
> Thanks for the question. We will clarify our contributions by specifying that we focus on MSE loss in our study of supervised learning.
>
> [a] Hui, Like, and Mikhail Belkin. "Evaluation of neural architectures trained with square loss vs cross-entropy in classification tasks." ICLR 2021
>
> > 2. In Equation 7, where the augmentation distribution is formalized, is my understanding correct that the directions that are unimportant to the task are augmented more, i.e., there is more noise added, as alpha grows, in the directions with $\lambda_i^{\mathbf{Γ}} = 0$. Is this correct?
>
> Yes, your intuition is exactly right, the directions that are unimportant to the task (which we refer to as *irrelevant noise features*) are indeed augmented more as $\alpha$ increases. Concretely, these correspond to components with $\lambda_i^{\mathbf{Γ}} > 0$ (rather than $\lambda_i^{\mathbf{Γ}} = 0$), since $\mathbf{Γ}$ denotes the covariance of the noisy, task irrelevant signal components. By increasing $\alpha$, our augmentation process injects proportionally more noise into these irrelevant directions, thereby encouraging the model to become invariant to them.
>
> Thank you for pointing this out, we will clarify this intuition in the revised manuscript.
>
> > 3. As a minor element and more of a suggestion, I believe the proofs would benefit more people looking to understand theoretical research in SSL if some of the identities, e.g., regarding matrix traces and norms, were expanded more in the first few derivations. This greatly helps the readability of the more technical material.
>
> Thank you very much for this helpful suggestion. We will expand the derivations to guide readers more smoothly through the proof. We’ll also include a comprehensive glossary of notation to further enhance clarity.
>
> We would be happy to address any remaining questions if needed.

---

> > ### Comment · Reviewer_FFZ4 · 2025-08-08
> > **reply to author rebuttal**
> >
> > I sincerely thank the authors for their explanations regarding the loss function used for the theoretical analysis, the notation and ideas behind the augmentation distribution and also the notation. I have no further questions

---

### Official Review · Reviewer_CuYw · 2025-07-04

**Clarity:** 4
**Significance:** 3
**Originality:** 4
**Rating:** 5
**Confidence:** 4

**Summary:**

This paper studies the trade-offs between two major self-supervised learning paradigms (reconstruction-based and joint-embedding-based methods) by analysing how these methods respond to noise and data augmentation. The authors derive closed-form solutions for linear models under both objectives, enabling a precise comparison of how augmentation alignment with irrelevant (noisy) input features affects learning. A central finding is that joint-embedding methods are more robust in high-noise scenarios, as they require weaker alignment conditions to achieve good performance. Conversely, reconstruction-based methods are shown to be preferable in low-noise settings, where they benefit from the ability to directly reconstruct relevant features without needing heavy augmentation tuning. These insights are validated across synthetic datasets, MNIST-style tasks, and ImageNet experiments. The paper provides useful theoretical grounding for an important design choice in SSL.

**Questions:**

1. In Appendix Figure 3, the probing error for the reconstruction-based method under high alignment ($\alpha = 10$) is not consistently lower than that of $\alpha = 1$ or even $\alpha = 0.1$. This appears to deviate from Proposition 4.3, which suggests that performance should improve with stronger alignment. Could the authors elaborate on whether this is due to finite sample effects, model instability under large augmentations, or something else? A brief discussion of this discrepancy and its implications would be valuable.

2. Could the authors clarify what latent dimensionality (i.e., value of $k$) was used in the synthetic experiments? While the theoretical derivations describe closed-form solutions in terms of the top‑$k$ components, I couldn't find an explicit mention of the actual value chosen. Since this dimensionality affects the probing performance and the comparison between methods, it would be helpful to know how $k$ was selected and whether it was consistent across settings.

3. The experimental settings for the reconstruction-based method do not mention the use of masking, which is a common component in modern reconstruction-based SSL approaches (e.g., MAE). Could the authors clarify whether any form of input masking was used in their reconstruction experiments? If not, do they expect their conclusions about the comparative performance of reconstruction vs. joint-embedding methods to hold if masking were incorporated?

**Ethical Concerns:**

["NO or VERY MINOR ethics concerns only"]

**Final Justification:**

I maintain my score (5, accept). This paper provides a clear and rigorous theoretical comparison of reconstruction-based and joint-embedding SSL methods, offering valuable insights into the role of augmentation alignment and noise. My initial questions regarding the probing error behaviour in Figure 3, the choice of latent dimensionality in synthetic experiments, and the use of masking were all addressed in the rebuttal.
Overall, the paper makes a meaningful theoretical contribution and offers insights that are likely to guide future SSL research and design.

**Limitations:**

yes

**Paper Formatting Concerns:**

no formatting concerns

**Quality:**

3

**Strengths And Weaknesses:**

Quality:
The theoretical results are clearly stated, with assumptions that are reasonable in the linear setting. Proofs in the appendix are detailed and rigorous. On the experimental side, the setup is comprehensive, covering both synthetic and real-world data, and includes both linear and deep model evaluations. One area that could benefit from further discussion is the reconstruction-based method's behaviour in Figure 3. Its performance does not always improve with stronger augmentation alignment (higher $\alpha$), which seems at odds with the theoretical predictions. This may reflect finite-sample effects or practical limitations, and would be worth addressing more directly.

Clarity:
The writing is clear, and the paper is logically organised. The theoretical development is presented in a way that is accessible to readers familiar with SSL, and the motivation for each component is well articulated. Figures are thoughtfully chosen and support the key messages. The appendix is well structured and easy to navigate.

Significance:
The paper offers a thoughtful theoretical perspective on when to prefer joint-embedding versus reconstruction-based SSL methods. It adds clarity to a design choice that is often made based on empirical benchmarks or intuition. The analysis of how noise and augmentation alignment influence performance is particularly insightful and contributes to a deeper understanding of SSL behaviour. That said, in real-world scenarios, it may not always be clear whether one is operating in a low- or high-noise regime, which could limit the immediate practical applicability of the guidance. Nonetheless, the theoretical framework is likely to inform future research and help shape more principled approaches to SSL model development.

Originality: A key strength of the paper is its analytical treatment of the impact of augmentations. By explicitly modelling the augmentation distribution, the authors derive closed-form solutions that make it possible to analyse how SSL objectives respond to alignment between irrelevant features and augmentation. This leads to actionable insights, such as when reconstruction-based SSL is preferable and when joint-embedding is more robust. This kind of augmentation-aware analysis is novel in the SSL literature and adds a useful new angle to our theoretical understanding of self-supervised learning.

---

> ### Author Rebuttal · Authors · 2025-07-25
>
> We would like to thank the reviewer for the insightful review and very positive feedbacks.
>
> > 1. In Appendix Figure 3, the probing error for the reconstruction-based method under high alignment [...]
>
> This is indeed a consistent and interesting point to discuss.
>
> We believe this discrepancy stems from finite‑sample effects. By the multivariate Central Limit Theorem, the variance of the empirical noise‑covariance spectrum decays like $1/\sqrt{n}$. In low‑sample regimes, this high variance causes the spectrum to deviate from the true noise covariance $\mathbf{\Gamma}$, thus worsening probing error. Because the excess variance scales with $\alpha^2$, its impact is greatest at large $\alpha$. As $n$ increases, these fluctuations vanish, and strong alignment consistently outperforms weaker settings, as predicted by Proposition 4.3.
>
> Importantly, we do not observe this behavior with joint embeddings: because both views undergo the same augmentation pipeline, they maintain identical variance levels for each $\alpha$, even in small‑sample scenarios.
>
> We will incorporate this clarification into the revised manuscript. Thank you for your feedback.
>
>
> > 2. Could the authors clarify what latent dimensionality (i.e., value of k) was used in the synthetic experiments? [...]
>
> We apologize for the lack of clarity. In the synthetic experiments, we set the latent dimensionality to $k=50$ informative components (the top 50 principal components of the input data) plus 50 pure‐noise dimensions. Although this setup is described in Appendix C, we will state it explicitly in the main text.
>
> We selected $k=50$ heuristically, and we observed that the qualitative patterns hold for other choices of $k$ as well. In practice, the relative magnitudes of the data components matter more than the exact number of dimensions.
>
>
> > 3. The experimental settings for the reconstruction-based method do not mention the use of masking, which is a common component [...]
>
> In section 5.2, we indeed experimented with MAE that implements masking as data augmentation, as provided by the lightly SSL package : https://github.com/lightly-ai/lightly. We use the hyperparameters for ImageNet as provided in the original MAE paper [a]. Results are reported in Table 1 and show that MAE is much more sensible to data corruption than Joint-Embedding approaches, thus confirming the insights of Corollary 4.5.
>
> Moreover, if the reviewer is interested, we refer to our answer to reviewer ufsf for explanations on how the covariance operator of masking (as defined in [b]) can be leveraged to derive theoretical results similar to those obtained in the Gaussian case (section 4 in the paper). Exploring specific covariances related to known data augmentation functions, rather than generic ones, is a promising direction for future work.
>
> [a] He, Kaiming, et al. "Masked autoencoders are scalable vision learners." Proceedings of the IEEE/CVF conference on computer vision and pattern recognition. 2022.
>
> [b] Lin, Chi-Heng, et al. "The good, the bad and the ugly sides of data augmentation: An implicit spectral regularization perspective." JMLR 25.91 (2024): 1-85.
>
> We would be happy to address any remaining questions if needed.

---

> > ### Comment · Reviewer_CuYw · 2025-08-05
> >
> > I thank the authors for clarifications regarding finite-sample effects, latent dimensionality, and the use of masking. I have no further questions.

---

### Note · Authors · 2025-08-11

Dear AC and reviewers,

Thank you for the time and effort you put into reviewing our submission. Your feedback was very encouraging and has helped us improve the paper in many ways: emphasizing the importance of our work, clarifying technical points and assumptions, and making our derivations easier to follow. We will include all these improvements in the revised version.

We are truly grateful for your thoughtful input and support.

---

### Decision · Program_Chairs · 2025-09-17

**Decision:**

Accept (spotlight)

**Comment:**

All reviews of this work leaned toward acceptance, with two borderline accept recommendations [YR7n,ufsf], one accept recommendation [CuYw] and one strong accept recommendation [FFZ4].

The reviewers appreciated several aspects of the work:

+ The work was considered well motivated [YR7n] and the research question was considered practical and important [ufsf]
+ The theoretical perspective on joint embedding versus reconstruction-based SSL was appreciated [CuYw,FFZ4]
+ The theoretical results were appreciated [YR7n], the theory was considered clean [ufsf] and the results were considered clearly stated [CuYw] and derived [FFZ4] and were considered to align with intuition [FFZ4]
+ The results on noise and augmentation regimes was considered important [FFZ4]
+ Analytical treatment of impact of augmentations was appreciated and considered novel [CuYw]
+ The assumptions were considered reasonable in the linear setting [CuYw]
+ The proofs were considered detailed and rigorous [CuYw]
+ The experiments were appreciated [YR7n,ufsf] and the experimental setup was considered comprehensive [CuYw]; another reviewer considered the experiments few but confirming the theoretical results [FFZ4]
+ The writing was considered clear [CuYw]


However, several concerns were also raised, such as concerns on practical guidance for practitioners, and reliance of the proofs on various assumptions. Authors addressed several of the concerns in the rebuttal stage. In detail:

- More discussion of the reconstruction-based method's behaviour in Figure 3 was desired [CuYw] and similarly explanation for behaviour of the model with string alignment in Figure 2 bottom right [YR7n]; authors speculated this arises from finite sample effects.
- Practical applicability of the guidance was considered potentially limited due to uncertainty of whether one is in a low or high noise regime [CuYw]
- The guidance for practitioners was considered thin [ufsf] and lacking demonstration of how to estimate alignment or noise strength on real unlabelled data [ufsf]; authors claimed estimation is beyond the scope of the work.
- Proof reliance on additive Gaussian noise and scaled augmentations was criticised [ufsf] and generalisation to real augmentations was unclear [ufsf]; authors argued Gaussian modelling was a natural second-order approximation.
- Assumption or orthogonal directions for noise and signal was considered to rarely hold for natural images or text [ufsf] and intuition for settings where they overlap in the same subspace was desired [ufsf]; authors noted some of their experiments were in such settings.
- A question was raised about the continuous labels assumption and its impact on SSL model downstream performance on categorical matching tasks [FFZ4]; authors provided some discussion.
- A clarification on augmentation of unimportant directions was requested [FFZ4]; authors clarified it.
- The meaning of good alignment of noise and augmentation was considered unclear [YR7n];
- Goodness of linear models as proxies for non-linear ones was questioned [YR7n]; authors argued linear refers only to dependence on parameters and not to representation power.
- Controlling for unaligned noise was questioned [YR7n]; authors provided some discussion.
- Some clarifications for proofs were requested [YR7n]; authors provided detail.
- Clarification of the latent dimensionality in the synthetic experiments was requested [CuYw]; authors clarified it.
- Clarification of masking was requested [CuYw]; authors discussed where it was used.
- Improvements on readability of some proofs and their identities was suggested [FFZ4]; authors agreed to expand the derivations.
- Lack of discussion of limitations was criticised [ufsf]; authors promised to more explicitly discuss key assumptions and scope.

After the rebuttal reviewers [CuYw,FFZ4,YR7n] had no further questions. It was unclear whether all concerns of [ufsf] were addressed but at least some seemed to be.

Overall, it seems even though some theoretical concerns mat remain, the work has clear contributions that were appreciated and can have at least theoretical impact even if not always practical applicability. Overall, it seems the contributions of the work are interesting enough that the work could be presented NeurIPS.